# Constraint-Driven Explanations of Black-Box ML Models

## Abstract

Modern machine learning techniques have enjoyed widespread success, but are plagued by lack of transparency in their decision making, which has led to the emergence of the field of explainable AI. One popular approach called LIME, seeks to explain an opaque model's behavior, by training a surrogate interpretable model to be locally faithful on perturbed instances. Despite being model-agnostic and easy-to-use, it is known that LIME's explanations can be unstable and are susceptible to adversarial attacks as a result of Out-Of-Distribution (OOD) sampling. The quality of explanations is also calculated heuristically, and lacks a strong theoretical foundation. In spite of numerous attempts to remedy some of these issues, making the LIME framework more trustworthy and reliable remains an open problem.

In this work, we demonstrate that the OOD sampling problem stems from rigidity of the perturbation procedure. To resolve this issue, we propose a theoretically sound framework based on uniform sampling of user-defined subspaces. Through logical constraints, we afford the end-user the flexibility to delineate the precise subspace of the input domain to be explained. This not only helps mitigate the problem of OOD sampling, but also allow experts to drill down and uncover bugs and biases hidden deep inside the model. For testing the quality of generated explanations, we develop an efficient estimation algorithm that is able to certifiably measure the true value of metrics such as fidelity up to any desired degree of accuracy, which can help in building trust in the generated explanations. Our framework called CLIME can be applied to any ML model, and extensive experiments demonstrate its versatility on real-world problems.

## 1 Introduction

Advances in Machine Learning (ML) in the last decade have resulted in new applications to safety-critical and human-sensitive domains such as driver-less cars, health, finance, education and the like (c.f. (Lipton, 2018)). In order to build trust in automated ML decision processes, it is not sufficient to only verify properties of the model; the concerned human authority also needs to understand the reasoning behind predictions (DARPA, 2016; Goodman & Flaxman, 2017; Lipton, 2018). Highly successful models such as Neural Networks and ensembles, however, are often complex, and even experts find it hard to decipher their inner workings. Such opaque decision processes are unacceptable for safety-critical domains where a wrong decision can have serious consequences. This has led to the emergence of the field of eXplainable AI (XAI), which targets development of both naturally interpretable models (e.g. decision trees, lists, or sets) (Hu et al., 2019; Angelino et al., 2018; Rudin, 2019; Avellaneda, 2020) as well as post-hoc explanations for opaque models (Ribeiro et al., 2016; Lundberg & Lee, 2017). Although interpretable models have been gaining traction, state-of-the-art approaches in most domains are uninterpretable and therefore necessitate post-hoc explanations.

One popular post-hoc approach is the Locally Interpretable Model-agnostic Explanation (LIME) framework (Ribeiro et al., 2016), which seeks to explain individual predictions by capturing the local behaviour of the opaque model in an approximate but interpretable way. To do so, it trains a surrogate interpretable classifier to be faithful to the opaque model in a small neighborhood around a user-provided data instance. Specifically, the given data instance is perturbed to generate a synthetic data set on which an interpretable classifier such as a linear model is trained with the objective of having high fidelity to the original model. The human decision-maker can inspect the coefficients of the linear model to understand which features contributed positively or negatively to the prediction.

LIME is model-agnostic as it treats the opaque model as a black-box. It is applicable to a wide variety of data such as text, tabular and image, and is easy to use. Nevertheless, it suffers from sensitivity to out-of-distribution sampling, which undermines trust in the explanations and makes them susceptible to adversarial attacks (Slack et al., 2020). Post-hoc explanation techniques are increasingly being used by ML experts to debug their models. In this setting, LIME does not provide the user with the flexibility to refine explanations by drilling down to bug-prone corners of the input domain. Further, LIME has limited capabilities for measuring explanation quality accurately. For instance, checking whether the generated explanation is faithful to the original model can only be done on a heuristically defined number of samples, which can be unreliable. While later works such as Anchor (Ribeiro et al., 2018) and SHAP (Lundberg & Lee, 2017) mitigate some of these issues, LIME remains a popular framework and making it more robust and reliable remains an open problem.

Towards this goal, we first demonstrate that LIME's rigid perturbation procedure is at the root of many of these problems, due to its inability to capture the original data distribution. We craft concrete examples which clearly show LIME generating misleading explanations due to OOD sampling. Further, LIME affords the end-user limited control over the sub-space to be explained; since the notions of 'locality' and 'neighborhood' are defined implicitly by LIME's perturbation procedure. XAI, however, is inherently human-centric and one-size-fits-all approaches are not powerful enough to handle all user-needs and application domains (Sokol et al., 2019). Instead, we propose to generalize the LIME framework by letting the user define the specific subspace of the input domain to be explained, through the use of logical constraints. Boolean constraints can capture a wide variety of distributions and recent advances in SAT solving and hashing technology have enabled fast solution sampling from large complex formulas (Soos et al., 2020). Making LIME's neighborhood generation flexible in this way resolves both limitations of LIME. First, our approach helps in mitigating the problem of OOD sampling. Second, it allows the user to drill down and refine explanations. For instance, it might be not sufficient for doctors to simply know which test results contributed to a model's prediction of cancer; it is also important to understand the prediction in the context of the patient's specific age group, ethnicity etc. Such requirements can be naturally represented as constraints. In the same vein, constraints can also be used to zoom in to bug-prone corners of the input space to uncover potential problems in the model. This is especially useful for model debugging which is a recent direction for ML research (Kang et al., 2020). Letting users define sub-spaces to be explained also necessitates a theoretically grounded method of measuring explanation quality, as a poor quality score can indicate to the user, the need for refining constraints. Existing works compute these metrics heuristically without formal guarantees of accuracy. In this light, we propose a theoretical framework and an efficient estimation algorithm that enables measurement of the true value of metrics like fidelity, up to any desired accuracy, in a model-agnostic way. Through extensive experiments, we demonstrate the scalability of our estimation framework, as well as applications of CLIME to real world problems such as uncovering model biases and detecting adversarial attacks.

In summary, our contributions are as follows:

1. Framework for precisely crafting explanations for specific subspaces of the input domain through logical constraints

2. A theoretical framework and an efficient algorithm for estimating the 'true' value of metrics like fidelity up to any desired accuracy

3. Empirical study which demonstrates the efficacy of constraints in
   - Mitigating problem of OOD sampling
   - Detecting adversarial attacks
   - Zooming in and refining explanations for uncovering hidden biases

## 2 PRELIMINARIES

**Problem formulation.** We follow notations from Ribeiro et al. (2016). Let $D = (X, y) = \{(x^1, y^1), (x^2, y^2), \ldots, (x^n, y^n)\}$ denote the input dataset from some distribution $\mathcal{D}$ where , $x^i \in \mathbb{R}^d$ is a vector that captures the feature values of the $i$th sample, and $y^i \in \{\mathcal{C}_0, \mathcal{C}_1\}$ is the corresponding class label[1]. We use subscripts, i.e. $x_j$, to denote the $j^{th}$ feature of the vector $x$.

---

[1] We focus on binary classification; extension to multi-class classification follows by 1-vs-rest approach

Let $f : \mathcal{R}^d \to [0, 1]$ denote the opaque classifier that takes a data point $x^i$ as input and returns the probability of $x^i$ belonging to $\mathcal{C}_1$. We assume that an instance $x$ is assigned label $l_f(x) = \mathcal{C}_1$ if $f(x) \geq 0.5$ and $l_f(x) = \mathcal{C}_0$ otherwise. Given a classifier $f$, the task is to learn an explainer model $g$ such that $g$ mimics the behavior of $f$ in the neighborhood of some given point $x$. The type of classifier $g$ is chosen to be visually or textually interpretable, so that it can serve as an explanation for $f$'s behavior in the neighborhood of $x$.

**Overview of LIME.** LIME builds the explainer function $g$ on an 'interpretable domain' of inputs rather than the original domain. To do so, it maps original features (that can be continuous or categorical) to Boolean features. While $x \in \mathbb{R}^d$ represents an instance in the original domain, we use prime-notation, i.e. $x' \in \{0, 1\}^{d'}$ to represent an instance in the interpretable domain. Using Boolean features is a natural choice for 'interpretable domain', as we can understand explanations in terms of a presence/absence of a feature's value. Thus $g$ operates in the interpretable domain $\{0, 1\}^{d'}$.

LIME randomly perturbs the given instance $x$ to generate neighbors $z^1, z^2, \ldots$ with corresponding interpretable versions $z'^1, z'^2, \ldots$. The set of sampled neighbors i.e., the neighborhood of $x$, is denoted by $\mathcal{Z}$ ($\mathcal{Z}'$ in the interpretable space). Additionally, we denote the universe of all possible neighbors of $x$ (not just the ones sampled) by $\mathcal{U}^Z$ and $\mathcal{U}^{Z'}$.

Let the complexity of an explanation $g$ be denoted as $\Omega(g)$ (complexity of a linear model can be the number of non-zero weights), and let $\pi_x(z)$ denote the proximity measure between inputs $x$ and $z \in \mathcal{Z}$ ($\pi_x(z)$ can be defined using cosine or $L_2$ distance). The objective function for training $g$ is crafted to ensure that $g$: (1) approximates the behavior of $f$ accurately in the vicinity of $x$ where the proximity measure is high, and (2) achieves low complexity and is thereby interpretable. The explanation is obtained as $g^* = argmin_{g \in G} \, L(\pi_x, g, f) + \Omega(g)$ where $G$ is the set of all linear classifiers and the loss function $L$ is defined as: $L(f, h, \pi_x) = \sum_{z \in \mathcal{Z}, z' \in \mathcal{Z}'} [f(z) - g(z')]^2 \pi_x(z)$. These are heuristically defined functions. Intuitively, the loss function captures how unfaithful $g$ is to $f$ in the neighborhood of $x$.

**Boolean (logical) constraints and uniform sampling.** We use notation standard in the area of Boolean Satisfiability (SAT). A Boolean formula over $n$ variables $\phi : \{0, 1\}^n \to \{0, 1\}$ assigns a truth value $0/1$ or false/true to each of the $2^n$ assignments to it's variables and is constructed using logical operators like AND ($\wedge$), OR($\vee$), NOT ($\neg$), XOR ($\oplus$) etc. An assignment of truth values to variables denoted $s \in \{0, 1\}^n$ is said to satisfy $\phi$ (denoted $s \models \phi$) iff $\phi(s) = 1$. The total number of assignments that satisfy $\phi$ is denoted as $\#\phi = \sum_{s \in \{0,1\}^n} \phi(s)$. An algorithm is said to be a (perfectly) uniform sampler if it takes as input an arbitrary formula $\phi$ and returns an assignment $s^*$ such that $\forall s \models \phi$, we have $Pr[s^* = s] = \frac{1}{\#\phi}$. An almost-uniform sampler is an algorithm that takes, along with $\phi$, a parameter $\varepsilon > 0$ as input and returns $s^*$ such that $\forall s \models \phi$, we have $\frac{1}{(1+\varepsilon)\#\phi} \leq Pr[s^* = s] \leq \frac{1+\varepsilon}{\#\phi}$. The tools WAPS (Gupta et al., 2019) and Unigen3 (Soos et al., 2020) are state-of-the-art perfectly uniform and almost-uniform samplers respectively.

**Fidelity.** The fidelity $\rho$ of the explainer model $g$ is a measure of how closely the predictions made by $g$ match those of $f$ in the neighborhood around the given point $x$ (Ribeiro et al., 2016). It is calculated as the fraction of the sampled neighbors where the output class of $f$ and $g$ agree, i.e.

$$\widehat{\rho} = \frac{\sum_{z \in \mathcal{Z}} \mathcal{I}[l_f(z) == l_g(z')]}{|\mathcal{Z}|} \tag{1}$$

where $\mathcal{I}$ is the indicator function.

## 3 ANALYSIS OF LIME

**LIME sampling procedure.** LIME generates the neighborhood of the data point under explanation differently for text, image and tabular datasets. We consider tabular data here (see Appendix B for omitted details, examples, etc.). For tabular data, LIME samples points $z^i \in \mathcal{Z}$ in the neighborhood of $x$ (continuous features are discretized) and then maps them to the interpretable (binary) domain as $z'^i \in \mathcal{Z}'$. The distance function $\pi$ is defined on the binarized space to measure distance between the original $x'$ and sampled instances $z'$. Given $(\mathcal{Z}', \mathcal{Z}, \pi)$ LIME builds a linear regression model $g$ to fit

these data, where the distance measure $\pi$ defines weights of samples.

$$g(z') = \sum_{i=1}^{T} c_i z_i' + d. \tag{2}$$

**Framework characteristics.** The underlying assumption of LIME is that it is possible to approximate the behavior of the black box classifier accurately around $x$, so that $g$ can faithfully mimic $f$ in $x$'s neighbourhood. To fulfill this assumption, LIME presumes that it is possible to produce a representation of the neighbourhood $\mathcal{Z}$ of $x$ that adheres to the true distribution of the data $\mathcal{D}$ (i.e. $\mathcal{Z} \sim \mathcal{D}$), through its fixed sampling procedure. However, this assumption is quite strong. Even if the classifier $f$ behaves linearly in the neighbourhood of $x$, the correct distribution of instances can vary depending on the problem being solved, which may not be accounted for by a fixed sampling procedure. Therefore, the key question is *does LIME's sampling procedure produce a neighbourhood $\mathcal{Z}$ s.t. $\mathcal{Z} \sim \mathcal{D}$?*. Another more practical question is *does OOD sampling lead to unreliable explanations?* In fact, Slack et al. (2020) already gamed LIME's out-of-distribution (OOD) samples using an additional scaffolded classifier that behaves exactly like $f$ on samples from the distribution and produces arbitrary decisions for other samples. We show that even without wrapping a classifier, LIME's explanations suffer dramatically from OOD sampling.

To do so, we introduce a simple metric that measures the severity of OOD sampling. The metric is based on the observation that if $\mathcal{Z} \sim \mathcal{D}$ (i.e. no OOD sampling occurs), then the fraction of instances with 'true' label $\mathcal{C}_1$ should be roughly the same for both $\mathcal{Z}$ and $\mathcal{D}$. Let $W_Q$ be the fraction of intances $q \in Q$ that have true label $l(q) = \mathcal{C}_1$, i.e. $W_Q = \sum_{q \in Q} \mathcal{I}(l(q) == \mathcal{C}_1)/|Q|$. Then if $\mathcal{Z} \sim \mathcal{D}$, we should have $W_\mathcal{Z}/W_\mathcal{D} \approx 1$. By construction of the binary space $\mathcal{Z}'$, we know that $W_\mathcal{Z} = W_{\mathcal{Z}'}$. Therefore, we should also get $W_{\mathcal{Z}'}/W_\mathcal{D} \approx 1$. For an explainer $\mathbb{E}$, we define

$$SQ(\mathbb{E}) = W_{\mathcal{Z}'}/W_\mathcal{D} \tag{3}$$

If the value of $SQ(\mathbb{E})$ is far from 1, it is indicative of OOD sampling. However, computing $SQ(\mathbb{E})$ directly is infeasible in practice, as we generally do not have access to the true labels of instances generated through sampling. Therefore we craft a few classification tasks where a perfect classifier $f$ can be trained such that $f(x) \geq 0.5$ for $\forall x, x \sim \mathcal{D}$ such that $x$ belongs to $\mathcal{C}_1$. This allows us to obtain the true labels for computing $W_{\mathcal{Z}'}$ and $W_\mathcal{D}$. We show such an example in Fig.1, called 'Hit or miss example'. In Fig.1, the squares $S_0$ and $S_1$ represent the areas of the input space that are in-distribution, while points outside the squares (white regions) are OOD. Intuitively, when perturbing the point with coordinates $(8, 8)$, for example, LIME's distribution-unaware sampling procedure is likely to generate a lot of OOD samples, as the neighborhood of $(8, 8)$ is largely OOD (white). In Example B.1 in Appendix B, we precisely define the classification task in this example. We found $SQ(LIME) = 0.75$ which is much less than 1, and clearly indicates that LIME's sampling procedure does not satisfy the main underlying assumption $\mathcal{Z} \sim \mathcal{D}$.

**Framework capabilities.** The other limitation of LIME is that the fixed perturbation procedure gives users very little flexibility in tailoring explanations to their needs. Users can only control the data point $x$ to be explained, which can be insufficient for different use-cases. It is natural to ask: Can we generate explanations using a specific subspace of inputs? Namely, can we formally define $\mathcal{Z}$ instead of having LIME define it for us? The need for customizing $\mathcal{Z}$ arises naturally in the process of refining explanations, or when debugging models. For example, regulators may want to know how insurance applications are decided for people with specific preexisting conditions. Traders may need to understand the reason behind a decision to sell, in the context of the prevailing market conditions. Answering such queries is challenging in the existing LIME setting as it requires sampling capabilities from a constrained subspace. In our proposed framework, the user can naturally enforce such requirements through constraints such as ordering relations, arithmetic expressions over bounded integers, Boolean expressions or any logical combinations of these.

## 4 GENERATING AND CERTIFYING CONSTRAINT-DRIVEN EXPLANATIONS

From the preceding discussion, it is clear that the limitations of the LIME framework that remain unaddressed are (a) OOD sampling, (b) ability to work in constrained spaces and (c) providing verifiable guarantees on fidelity. Towards the goal of remedying this, we first present our constraint-driven explanation generation framework called CLIME. Then, we show how we can obtain certifiably accurate measurements of an explanation's quality through a novel estimation algorithm.

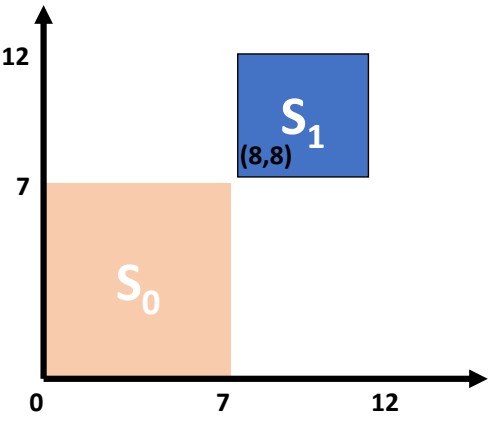

Figure 1: Hit or miss example

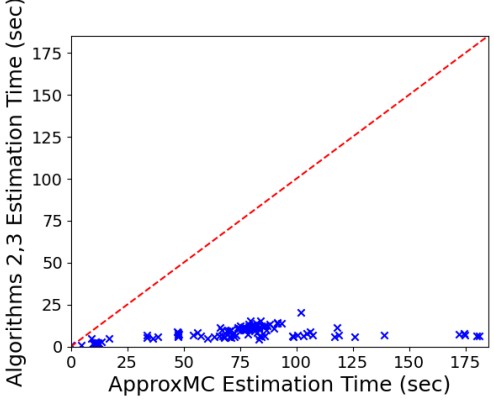

Figure 2: Scalability of Alg. 2,3 vs. ApproxMC

## 4.1 CLIME EXPLANATION FRAMEWORK

The CLIME framework generates explanations on instances sampled (almost) uniformly from user-defined subspaces which are defined through constraints. Boolean constraints are powerful enough to represent log-linear family of distributions (Chavira & Darwiche, 2008), yet allow fast sampling of solutions either uniformly or with a user-provided bias (Gupta et al., 2019), thanks to advances in SAT technology. In this work, we focus on explanations based on (almost) uniformly sampling solutions of constraints, but we note that the extension to biased (weighted) sampling is straightforward.

The pseudo-code of the constrained explanation framework is presented in Alg. 1. The key difference with respect to LIME is that along with the input instance $x$ CLIME also takes as input a Boolean formula $\phi$. The variables of $\phi$ are exactly the Boolean features of the interpretable domain $\mathcal{Z}'$. As an example, assume that $\phi$ represents the constraint that at least 'c' variables must be true for some user-defined 'c'. For image data this constraint enforces the requirement that at least 'c' superpixels must be 'on', while for text it forces at least 'c' words from $x$ to be present in each $z$. This blocks out very sparse data ensuring that only informative instances are used for training the explainer model.

We assume access to a procedure $getSamples$ that returns $N$ independent samples satisfying $\phi$. The algorithm takes as input a parameter $\varepsilon$, which represents the tolerance to deviation from perfectly-uniform sampling. If $\varepsilon = 0$, then the call to $getSamples$ in line 1 must be to a perfectly uniform sampler like WAPS (Gupta et al., 2019), otherwise an almost-uniform sampler like Unigen3 (Soos et al., 2020) suffices. The rest of the algorithm is similar to LIME: samples $z'^i$ are mapped back to the original domain, and the output of $f$ on each $z^i$ and the distance of each $z^i$ to $x$ are used for training $g$ in line 6, where at most $K$ coefficients are allowed to be non-zero to ensure interpretability.

In Appendix C, we consider two classes of crafted examples to demonstrate (a) undesired consequences of OOD sampling for LIME which are resolved in CLIME (Example C.1 and C.2) and (b) CLIME's capabilities to incorporate user-defined constraints (Example C.3).

Our first set of examples highlight that LIME does suffer from OOD sampling, leading to poor quality of explanations. In particular, $SQ(LIME)$ can be very low and LIME's explanations significantly differ from the true explanations. In contrast, giving the user the ability to define $\mathcal{Z}_\phi \sim \mathcal{D}$ using constraints as in CLIME, mitigates OOD sampling; $SQ(CLIME)$ is almost 1 and CLIME produces correct explanations (we recall that the ground truth is known in these examples).

Our second class of crafted examples illustrates the explanation debugging process, where we allow the user to specify the subspace $\mathcal{Z}_\phi$ to obtain an explanation from, e.g. to know how decisions are made for specific classes of people such as those with alcohol dependency. We show that CLIME can naturally handle these constrained user-spaces, which may be challenging for LIME.

**Algorithm 1** $explain\_with\_CLIME$

**Input:** $f$: Opaque classifier
$\phi$: Boolean constraints
$\varepsilon$: Tolerance $N$: Number of samples
$\pi_x$: Similarity kernel
$K$: Length of explanation
**Output:** $g$: Interpretable linear classifier
1: $\mathcal{Z}' \leftarrow getSamples(\phi, \varepsilon, N)$;
2: $Z \leftarrow \{\}$
3: **for** $z' \in \mathcal{Z}'$ **do**
4:     $Z \leftarrow Z \cup \{z', f(z), \pi_x(z)\}$
5: **end for**
6: $g \leftarrow$ K-LASSO$(Z, K)$

---

**Algorithm 2** $computeEstimate(\varepsilon, \delta, \gamma)$

**Input:** $\varepsilon$: Tolerance $\delta$: Confidence
$\gamma$: Threshold
**Output:** $\widehat{\rho}$: Estimate of $\rho$ (see Thm. 1)
1: **if** checkThreshold$(\varepsilon, \delta, \gamma)$ == True **then**
2:     **return** $\perp$
3: **end if**
4: $\widehat{\rho} \leftarrow AA'(0.4 * \varepsilon, 0.4 * \varepsilon, \delta)$
5: **return** $\widehat{\rho}$

---

**Algorithm 3** $checkThreshold(\varepsilon, \delta, \gamma)$

**Input:** $\varepsilon$: Tolerance
$\delta$: Confidence
$\gamma$: Threshold
**Output:** True whp if $\rho \leq \gamma - \varepsilon$
1: $\nu \leftarrow min(\varepsilon + \varepsilon^2/2 - \gamma\varepsilon/2, (\varepsilon - \gamma\varepsilon/2)/(1 + \varepsilon/2))$
2: $N \leftarrow \frac{1}{2\nu^2} \log(\frac{1}{\delta})$
3: $S \leftarrow getSamples(\phi, \varepsilon/2, N)$
4: $C \leftarrow 0$
5: **for** $s \in S$ **do**
6:     $c \leftarrow checkProperty(s)$
7:     $C \leftarrow C + c/N$
8: **end for**
9: **if** $C \leq \gamma$ **then**
10:     **return** True
11: **else**
12:     **return** False
13: **end if**

## 4.2 CERTIFYING EXPLANATION QUALITY

For increasing user trust, it is necessary to provide a measure of the quality of explanations generated. This is especially important for explanations of user defined sub-spaces, as it may be possible that no simple explanation exists for a large subspace, and further refinements to the constraints may be required to get a high quality explanation. The fidelity metric, as defined in Eqn. 1 measures quality in terms of the fraction of samples on which the prediction made by the explainer model, matches the prediction of the opaque model. A high fidelity score indicates that the explainer model is faithful to the opaque model in $\mathcal{Z}$. A low fidelity score can occur if, for instance, a linear classifier is used for explaining a highly non-linear decision boundary, making the explainer unreliable.

Two parameters influence the accuracy of the fidelity score: the number of samples in $\mathcal{Z}_\phi$ and the quality of these samples, i.e. their uniformity in $\mathcal{U}^\mathcal{Z}$. Both of these parameters were chosen heuristically in prior works including LIME, which makes the explanations harder to trust. Intuitively, a score measured on 10 samples will not be as stable as one measured on 10000 due to randomness inherent in any sampling procedure. Ideally one would want to compute the fidelity on *all* possible instances belonging to a user-defined subspace of inputs.

Note that Eqn. 1 computes an estimate or sample mean of the 'true' population mean defined by

$$\rho = \frac{\sum_{z \in \mathcal{U}^{\mathcal{Z}_\phi}} \mathcal{I}[l_f(z) == l_g(z')]}{|\mathcal{U}^{\mathcal{Z}_\phi}|} \tag{4}$$

This observation allows us to compute the estimate $\widehat{\rho}$ in theoretically grounded way, so as to statistically guarantee its closeness to $\rho$. Such guarantees can be indispensable in safety-critical areas such as healthcare, to show adherence to regulatory requirements for AI, for instance.

Computing $\rho$ exactly is usually intractable, as the constraint $\phi$ is likely to have tens of hundreds of variables and exponentially many solutions. Approximating $\rho$ can be faster, but requires formal guarantees to be meaningful. A good approximation of $\rho$ is one that is within user-defined tolerance of the true value with high confidence. Specifically, $\widehat{\rho}$ is a good approximation of $\rho$ if

$$\Pr[(1 - \varepsilon)\rho \leq \widehat{\rho} \leq (1 + \varepsilon)\rho] \geq (1 - \delta) \tag{5}$$

where $\varepsilon > 0$, $\delta > 0$ are user-defined tolerance and confidence.

To the best of our knowledge, no existing approach is directly applicable for finding a good approximation of $\rho$, in a model-agnostic way. The technique presented by (Narodytska et al., 2019), requires the opaque model to be encoded as a Boolean formula, severely limiting types of models that can be explained. On the other hand, algorithms based on Monte Carlo sampling such as the $AA$ algorithm by (Dagum et al., 2000), are known to be fast when $\rho$ is high, but require far too many samples when $\rho$ is low (Meel et al., 2019). They also require perfectly uniform samples, while it may only be feasible to generate almost-uniform samples from the universe $\mathcal{U}^{\mathcal{Z}_\phi}$.

In this section, we describe an efficient estimation algorithm based on (Dagum et al., 2000), that is able to work with almost-uniform samples and also terminates quickly if the quantity being approximated is small. Two key insights inform the design of our approach: we first observe that $\varepsilon$-almost uniform sampling can change the value of $\rho$ at most by a factor of $(1 + \varepsilon)$. Secondly, in typical scenarios, users are interested in two-sided bounds on fidelity (as given by Eqn. 5) only if it is high enough. If the fidelity is lower than some threshold, say 0.1, then it doesn't matter if it is 0.05 or 0.01, since the explanation will be unacceptable in either case. In other words, below a certain threshold, one-sided bounds suffice.

We present our estimation framework in Algs. 2 and 3. Our technique is abstract in the sense that it can estimate the density of samples satisfying any property (not just fidelity), in any given domain (not just Boolean), so long as it is possible to sample (almost) uniformly from the domain (encapsulated in the procedure $getSamples$). We assume access to a procedure $checkProperty$, that given a sample $s$, returns 1 if the property of interest is satisfied by $s$ and 0 otherwise. In practice, for measuring fidelity of an explainer model on a subspace defined by some $\phi$, $getSamples$ can invoke WAPS or Unigen3 on $\phi$, while $checkProperty$ is simply the indicator function $\mathcal{I}[l_f(z) == l_g(z')]$.

Procedure $computeEstimate$ first invokes $checkThreshold$ which returns True with high probability $(1 - \delta)$ if the population mean is less than the prescribed threshold $\gamma$ with tolerance $\varepsilon$. If the check fails, then $computeEstimate$ makes a call to procedure $AA'$ (line 4), which is an adaptation of the algorithm by (Dagum et al., 2000) that provides guarantees similar to Eqn. 5 with almost-uniform samples (see Appendix D.1). Theorem 1 captures the guarantees and the behavior of the framework.

**Theorem 1.** *If $\rho \leq \gamma - \varepsilon$, then computeEstimate returns $\perp$ with high probability (i.e. at least $1 - \delta$). If $\rho \geq \gamma + \varepsilon$, w.h.p., it returns an estimate $\widehat{\rho}$ such that* $\Pr[(1-\varepsilon)\rho \leq \widehat{\rho} \leq (1+\varepsilon)\rho] \geq (1-\delta)$.

The benefit of leveraging the algorithm by (Dagum et al., 2000) is that it was proven to have close-to-optimal sample complexity. We demonstrate that this yields a fast certification tool in practice. We implemented and ran Alg. 2, 3 on 150 benchmarks used in Narodytska et al. (2019). The benchmarks are CNF formulas that encode the Anchor (Ribeiro et al., 2018) explanations of Binarized Neural Networks trained on Adult, Recidivism and Lending datasets. The true fidelity of an explanation can be computed from the count of the number of solutions of the corresponding formula. We compared the running-time of our tool to the time taken by the approach of (Narodytska et al., 2019), which utilizes the state-of-the-art approximate model-counting tool called ApproxMC (Soos et al., 2020). Note that both ApproxMC and our tool provide the same probabilistic guarantees on the returned estimate. The results are shown as a scatter-plot in Fig. 2. The x-coordinate of a point in blue represents the time taken by ApproxMC on a benchmark, while the y-coordinate represents the time taken by our approach. As all the points are far below the diagonal dotted red line, we can infer that ApproxMC takes significantly longer than our tool to compute the same estimate. In fact, on average (geometric mean), our tool is $7.5\times$ faster than ApproxMC. It is clear from Fig. 2 that our algorithm scales far better than the alternative, despite being more general and model-agnostic. We also experimentally compared the scalability of our tool to ApproxMC for different values of input tolerance $\varepsilon$ and confidence $\delta$. We found that our tool scales significantly better than ApproxMC for tighter tolerance and confidence values. Thus, our experiments demonstrate that our tool is efficient in practice. We provide more details and results in Appendix D.4.

## 5 EXPERIMENTS

We experimentally demonstrate how CLIME can be used for obtaining more meaningful explanations on synthetic and real-world datasets, i.e. *adult* (Kohavi, 1996) and *recidivism* (Schmidt & Witte, 1988). We also evaluate its ability to detect adversarial attacks. We focus on tabular data, but emphasize that CLIME is readily applicable to text and image data as well. Here we present results for recidivism (see extended version of results in Appendix E).

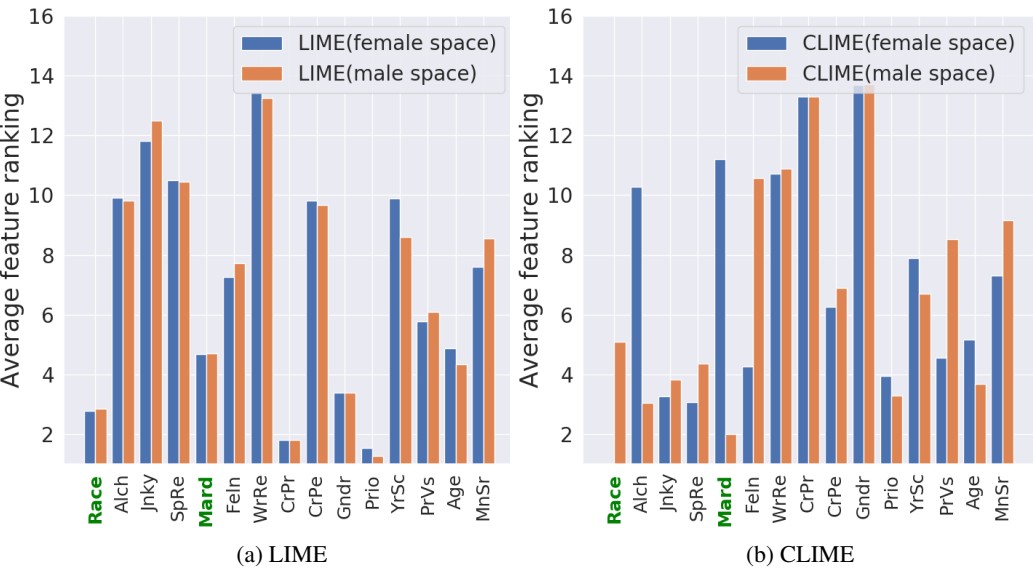

(a) LIME

(b) CLIME

Figure 3: Explanations for the recidivism dataset on female and male spaces.

**Recidivism Dataset.** The *Recidivism* dataset was used to predict recidivism for individuals released from North Carolina prisons. We used the discretized version of dataset as in (Ribeiro et al., 2018). We trained a small Multilayer Perceptron model that gives 69.1% accuracy on the test set. We consider two constrained spaces:

- 'female space': (JUNKY OR ALCOHOL) AND (CRIMEPROPERTY) AND GENDER = FEMALE

- 'male space': (JUNKY OR ALCOHOL) AND (CRIMEPROPERTY) AND GENDER = MALE.

We compute explanations for 50 samples within the corresponding constrained space. Figure 3 shows results for LIME (a) and CLIME (b), respectively, for both constrained spaces. At each plot, the blue histogram shows results for 'female space' and the orange histogram shows results for 'male space', respectively. Each bar shows the average rank of a feature in explanations, e.g. if the height of the bar is 1 the corresponding feature is mostly top ranked in all explanations.

First, we observe that the 'Gender' ('Gndr') and 'Crimeproperty' ('CrPr') features are fixed in these sub-spaces. Hence, these features are always ranked last by CLIME demonstrating the correct behaviour (any feature with a constant value cannot influence the decision so it should have the zero coefficient in the explainer model $g$ defined by Eqn. 2). Note that LIME is not informative enough for distinguishing between these constrained spaces. However, results of CLIME are very interesting. We observe that 'Race' is one of the top feature to make decision for *female* prisoners (the 'Race' blue bar has a small height) and only 4th (on average) for *males*. On the other hand, whether a male prisoner is 'Married' ('Mard') make the most difference which is also an interesting potential bias that we reveal.

**Detecting Adversarial Attacks.** (Slack et al., 2020) presented a way to craft an adversarial attack that seeks to hide the biased predictions made by a 'bad' classifier (ex: one that decides credit card applications solely on a sensitive feature like race) from detection by post-hoc explainers like LIME. Given a biased classifier, their idea was to construct an adversarial classifier that outputs the biased classifier's prediction if an input instance is detected to be in-distribution, and outputs a prediction based on an innocuous feature if the input instance is OOD. When LIME is used for explaining a prediction made by the adversarial classifier, almost all of LIME's perturbed samples are detected to be OOD, resulting in the innocuous feature appearing as the top explanation. In this way, the sensitive feature used for deciding in-distribution instances is successfully hidden. The attack crucially relies on the ability to train a classifier to distinguish between LIME's perturbed samples and instances

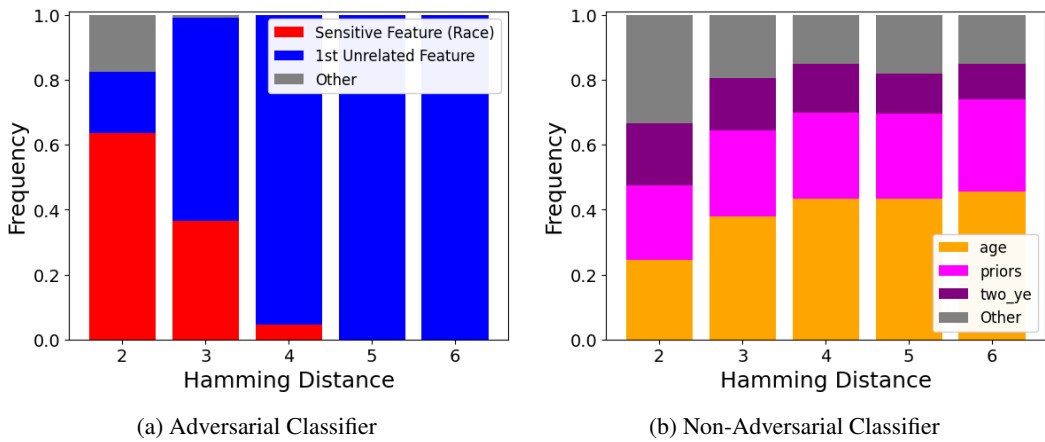

(a) Adversarial Classifier       (b) Non-Adversarial Classifier

Figure 4: Recidivism: Top CLIME explanation distribution vs. Hamming Distance

from the original distribution with high accuracy. In this experiment, we explore how CLIME can be used for detecting such attacks.

We hypothesized that for discrete data, the distinguishing classifier should have trouble differentiating between instances from the original dataset and instances with low Hamming distances to points in the original dataset. For a given Hamming distance $h$ and a set of instances $S$, it is possible to construct a formula $\phi^h$ such that the solutions of $\phi^h$ are exactly those instances that are at most at a distance $h$ from each instance in $S$. We took $S$ to be the set of 3 nearest neighbors of the instance being explained. If our hypothesis is correct, the adversarial classifier should have trouble hiding the sensitive features in the subspace $\mathcal{Z}_{\phi^h}$ for low values of $h$. Thus, sensitive features should appear in the top CLIME explanations when $h$ is small, while innocuous features should dominate for larger $h$.

To test our hypothesis, we generated 200 CLIME explanations using discretized versions of the three datasets used by (Slack et al., 2020). We computed the relative frequency of a feature appearing as the top explanation for different values of $h$. The results for the Recidivism dataset with 1 innocuous feature are depicted in Figure 4a (we defer results on other datasets and an extended discussion to Appendix E.3 for lack of space). We plot the hamming distance $h$ on the X-axis and the frequency of seeing a feature as the top explanation on the Y-axis. We see the sensitive feature 'race' appears as the top explanation for 60% of the instances explained when $h = 2$. The innocuous feature (1st Unrelated Feature), appears almost exclusively as the top explanation when $h \geq 5$. For comparison, we depict results on CLIME explanations for a non-adversarial classifier in Fig. 4b. We see that the relative frequency with which a feature appears as the top explanation changes very little for different hamming distances. Note that for LIME explanations (not shown), the innocuous feature appears as the top explanation for *all* of the instances explained. While it may be possible to craft even more sophisticated attacks, these results clearly demonstrates CLIME's ability to detect adversarial attacks that exploit OOD sampling.

## 6 CONCLUSIONS AND FUTURE WORK

We presented a model-agnostic explanation framework CLIME that is able to operate on constrained subspaces of inputs. We introduced a new metric for quantifying the severity of the OOD sampling problem and empirically demonstrated CLIME's ability to mitigate it. Additionally, our new estimation algorithm enables computation of an explanation's quality up to any desired accuracy. The need for making XAI more rigorous and evidence-based has been highlighted in the past (Doshi-Velez & Kim, 2017), and we believe our framework takes a concrete step in this direction. CLIME can also be readily extended in numerous ways. Helping the user with defining relevant subspaces by mining constraints from data is an interesting direction. Once sampling tools improve, richer constraint languages like SMT (Barrett & Tinelli, 2018) can provide even more flexibility. Construction of CLIME's explainer model can also incorporate Shapley values as in (Lundberg & Lee, 2017).

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

# A    RELATED WORK

Model explainability is one of the most important problems in machine learning. Therefore, there are a large number of recent surveys on the topic, e.g. Hoffman & Klein (2017); Hoffman et al. (2017); Lipton (2018); Adadi & Berrada (2018); Guidotti et al. (2019); Rudin (2019). To overview, we partition approaches to generate explanations for ML models into two groups based on whether they provide theoretical guarantees on the quality of the generated explanations.

**Explanations without theoretical guarantees.**    There were a number of approaches proposed to compute (model-agnostic) local explanations. We have overviewed LIME (Ribeiro et al., 2016) in Section 2. Anchor is a successor of LIME (Ribeiro et al., 2018). The main contribution of Anchor is to produce explanations that hold globally, for the entire distribution of inputs. SHAP (Lundberg & Lee, 2017) is another popular model-agnostic explainer to produce local explanations. Similar to other explainers, SHAP does not provide any theoretical justification for the sampling procedure. However, SHAP employs game theoretic principles to produce an explainable model. Our work focuses on model-agnostic, local explanations, however, we produce explanations with provable guarantees. CXPlain proposes to train an additional 'explanation model' to provide explanations for a given ML model (Schwab & Karlen, 2019). Learning of the explanation model involves an estimation of feature importance using a causal objective. The causal objective captures how input features cause a marginal improvement in the predictive performance of the ML model. In our work, we do not consider each feature individually and reason about the space of features as a whole. Moreover, our framework allows us to work with constrained spaces. Finally, works such as (Lakkaraju et al., 2019) provide limited capabilities for customizing global explanations by letting the user supply a set of features that the they deem important. Similar to (Björklund et al., 2019), they avoid sampling a neighbourhood around a given point by using original data points to construct an explainer. While avoiding sampling helps scalability, it also undermines applicability. For instance, dealing with user-defined constraints, as well as unbalanced or skewed input datasets can be problematic. In both cases, the input data may be too sparse to yield meaningful explanations. Recently, Lakkaraju et al. (2020) demonstrated that these explanations are less robust compared to LIME, for example.

Another line of work in on gradient-based explainers, for example, saliency maps (Zeiler & Fergus, 2014), Integrated Gradient (Sundararajan et al., 2017), DeepLIFT (Shrikumar et al., 2017). Gradient-based methods assume full knowledge about the ML model and, also, require these models to be differentiable. While these methods are very efficient, they do not provide theoretical guarantees on the produced explanations. On top of that these approaches are not model-agnostic.

**Explanations with theoretical guarantees.**    Recently, a formal approach to analyze explanation of ML models was proposed. If an ML model allows a formal representation in restricted fragment of the first order logic, then one can (a) define a formal notion of an explanation and (b) design an algorithm to produce these explanations (Shih et al., 2018; 2019; Ignatiev et al., 2019; Darwiche & Hirth, 2020; Darwiche, 2020). One of the formal approaches is built on powerful knowledge compilation techniques, e.g. (Shih et al., 2018; 2019). The other approach employees very efficient formal reasoners, like SAT, SMT or ILP solvers, as a part of explanation generation algorithms (Ignatiev et al., 2019; Narodytska et al., 2019). If the process of ML model compilation into a tractable structure is feasible then the first approach is very effective and allows the user to analyse the ML model efficiently. However, the compilation can be computationally expensive and resource demanding, so the second approach is more efficient in some applications. There are some limitations of these approaches. First, similar to gradient-based methods, they require full knowledge of the original ML model. Second, in practice, these approaches face scalability issues as reasoning about ML models formally is computationally expensive.

**Quality of the explanations.**    Finally, we consider a recent line of work on analysis of the quality of explanations. Ribeiro et al. (2018) proposed several heuristic measures to evaluate quality of explanations including fidelity and coverage, but do not provide a way to estimate the true value of these metrics. In (Ghorbani et al., 2019; Alvarez-Melis & Jaakkola, 2018), it was shown using perturbation-based methods that explanations are susceptible to adversarial attacks and lack robustness property. For example, Zhang et al. (2019) investigated several sources of uncertainty in LIME, like sampling variance in explaining a sample. The authors experimentally demonstrated that LIME often fails to capture the most important features locally. However, the paper does not propose a solution to

remedy identified issues. Moreover, Slack et al. (2020) showed that it is easy to fool an explainer, like LIME and SHAP, as we discussed in detail in Section 5. Narodytska et al. (2019) presented a technique for evaluation quality of explanations based on model counting, but their approach suffers from scalability issues (as shown in Sec. 4.2) and is only applicable to BNNs. Lakkaraju et al. (2020) proposed to use adversarial training (Madry et al., 2018) to improve robustness of the explanations. While the proposed approach improves robustness to adversarial attacks it cannot be easily extended to work in constraint environments and does not provide theoretical guarantees on the fidelity of the explanations. A related line of work on probabilistic verification of ML models has seen a surge in interest. (Albarghouthi et al., 2017) encoded the underlying model and fairness properties as formulas in SMT over real arithmentic, and relied on symbolic integration techniques. However, this approach is known not to scale, eg. it can only handle neural networks with a single hidden layer containing just three hidden units. (Bastani et al., 2019) present an alternative approach that uses Monte Carlo sampling and adaptive concentration inequalities. However, unlike Alg. 2, their method only returns a yes/no answer and does not provide a quantitative estimate. Further, their algorithm may fail to terminate on some inputs, and the sample complexity is not proven to be close-to-optimal.

## B   ANALYSIS OF LIME (ADDITIONAL MATERIALS)

### B.1   PERTURBATION PROCEDURE

LIME generates the neighborhood of the data point under explanation differently, depending on whether the given dataset is text, image or tabular. For text and images, LIME first maps a given point $x \in \mathcal{R}^d$ to $x' \in \{0,1\}^{d'}$ and then samples $z' \in \{0,1\}^{d'}$ in the neighborhood of $x'$. For tabular data, LIME first samples points $z^i$ in the neighborhood of $x$ and then maps them to the interpretable domain as $z'^i$. The distance function is also defined on the binarized space instead of the original space.

*Text and Images* For text and image data, the given point $x$ is first mapped to the Boolean domain. For text, the Boolean domain is simply the presence or absence of each word in $x$, while for images, $x$ is first divided into superpixels, and the Boolean domain is represented by the presence or absence of each superpixel. Points $z'$ in the neighborhood of $x'$ are sampled by flipping a few 1s to 0s where both the number and positions of 1s to flip are chosen randomly. Each $z'^i$ is mapped back to the original domain to obtain $z^i$ which are then input to $f$ for obtaining the 'true' labels. Interestingly, LIME's sampling procedure for text and images does not depend on the training set which is not the case for tabular data.

*Tabular* Tabular data can consist of both continuous and categorical features. Although LIME can work with continuous data natively, the recommended way is to perform quantization first using statistics of the original dataset. We denote $\mathcal{I}^j = \cup_{k=1}^{M_j}\{I_k^j\}$ the set of all intervals that we obtain due to discretization of the $j$th feature. To construct a neighborhood $\mathcal{Z}$ LIME uniformly samples features values from each interval in $\mathcal{I}^i$ for the $i$th feature. For categorical features, LIME builds a distribution of feature values and samples values for each feature in proportion to their frequencies.

*Construction of an interpretable domain.* Next, we consider how an interpretable binary domain is produced. For each sample $z$ in the neighborhood $\mathcal{Z}$, the corresponding $z'$ is constructed as

$$\text{Discretized features: } z'_j := bin(z_j) = \begin{cases} 1, & \text{if } \left(z_j \in I_k^j\right) \wedge \left(x_j \in I_k^j\right) \\ 0, & \text{otherwise} \end{cases} \tag{6}$$

$$\text{Categorical features: } z'_j := bin(z_j) = \begin{cases} 1, & \text{if } z'_j = x_j \\ 0, & \text{otherwise} \end{cases} \tag{7}$$

We denote $bin(z)$ the mapping function from the original feature vector $z$ to a binary vector $z'$, $z' = bin(z)$. Thus $\mathcal{Z}' = bin(\mathcal{Z})$. As this mapping is many-to-one, we denote *pre-image*$(z')$ the set of samples that are mapped to $z'$:

$$\textit{pre-image}_{\mathcal{Z}}(z') = \{z | bin(z) = z', z \in \mathcal{Z}\}. \tag{8}$$

*Interpretation of an explanation.* We recall the meaning of explanation for a vector $z$ given a linear regression model $g$ defined by Eqn. 2. Suppose the $i$th coefficient is great than zero, $c_i > 0$. Then the value $c_i$ is interpreted as a contribution toward the class 1. Namely, if the $i$th feature $f_i$ is not equal to $z_i$ then on average the value of $g(z)$ decreases by $c_i$. To see why it is so, we consider the binary space $z' = bin(z)$. If $f_i \neq z_i$ then $z'_i = 0$. Hence, contribution of the term $c_i z'_i$ to the value of $g(z')$ is decreased by $c_i$. Similarly, if $c_i < 0$ then the value $c_i$ is interpreted as a contribution to the class 0. If $f_i \neq z_i$ than the value of $g(z')$ is increased by $c_i$.

## B.2 Discussion on the quality of the perturbation procedure

The quality of explanations depends on the closeness of samples $\mathcal{Z}$ to samples from the original distribution $\mathcal{D}$. In Section 3, we introduce a simple metric $SQ(\mathbb{E}) = W_{\mathcal{Z}'}/W_{\mathcal{D}}$ (see Eqn. 3) to measure the quality of $\mathcal{Z}$. Please note that the $SQ$ metric might look similar to the fidelity metric that is defined in Eqn. 1. However, these metrics are different. $SQ$ measures quality of the sampling procedure based on the behavior of an ideal classifier on the generated samples, while $\widehat{\rho}$ evaluates the closeness of two models $g$ and $f$.

We recall that if the sampling procedure is correct then $SQ(\mathbb{E})$ should be close to 1 for a perfect classifier. While we can always experimentally measure $W_{\mathcal{Z}'}$, the value $W_{\mathcal{D}}$ cannot be compute in general. However, we will use few crafted classification tasks where $W_{\mathcal{D}}$ can be computed exactly to highlight the difference in behaviour of LIME and our proposed procedure.

The following crafted example is designed to highlight that OOD sampling occurs and our $SQ(\mathbb{E})$ metric is able to detect it. Moreover, we show that the quality as explanations suffer due to the OOD sampling.

**Example B.1** ('Hit or miss'). *We define the classification problem to illustrate issues with the OOD sampling. This problem models a situation when the classifier relies on seeing specific feature values in the input sample to make its decision. As we mentioned above, the reason we work with a crafted problem is that we can compute the value $W_{\mathcal{D}}$ exactly, which is not possible for real world datasets.*

***Definition of a task.*** *We consider two areas $S_0$ and $S_1$ on the 2D plane (see Figure 1(a)). A sample $x$, $x \sim \mathcal{D}$, represents integer coordinates of three points $p_1, p_2$ and $p_3$ drawn from $S_0 \cup S_1$. Hence, we have six features vector $x = [x_1, x_2, \ldots, x_6]$, where $p_i = (x_{2i-1}, x_{2i})$, $i = 1, 2, 3$, is a 2D point, or $x = [p_1, p_2, p_3] = [(x_1, x_2), (x_3, x_4), (x_5, x_6)]$. If there is $p_i \in x$ s.t. $p_i \notin S_0 \cap S_1$ then $x$ is an out-of-distribution sample. For example, $[(7, 7), (7, 7), (10, 10)]$ is an in-distribution sample, while $[(1, 10), (7, 7), (10, 10)]$ is an out-of-distribution sample as the first point $(1, 10) \notin S_0 \cap S_1$.*

*Given a sample $x \sim \mathcal{D}$, we say that $x$ belongs to $C_1$ iff there exists $i$ s.t. $p_i \in S_1$, $i \in [1, 3]$, and belongs to $C_0$ otherwise. For example, $[(7, 7), (7, 7), (10, 10)]$ belongs to $C_1$ as the third point $(10, 10) \in S_1$. So, if one of three points in $x$ hits $S_1$ then $x$ is classified as $C_1$. In this example, an explanation for a sample $x$ is well-defined as points in the $S_0$ (resp. $S_1$) area contribute to the $C_0$ (resp. $C_1$) class, respectively.*

***LIME performance.*** *We train an almost perfect neural network-based classifier for this simple example (See more in Section E.1). We compute our metric $SQ$ for LIME and get that $SQ(LIME) = 0.75$ which is less than 1 hinting that the sampling procedure is biased.*

*To see how $SQ$ affects explanations, we generate LIME explanations for a samples $x = [(7, 7), (7, 7), (7, 7)]$ that is close to the corner $(8, 8)$. As all points in $x$ are in $S_0$, $x$ belongs to class $C_0$. Table 1 shows explanations for this point in the column 'Balanced dataset'. The first row shows explanations produced by LIME. The red horizontal bar indicates that the corresponding feature has a negative coefficient in $g$ and it contributes toward $C_0$. Symmetrically, the green bar indicates that the feature contributes toward $C_1$. We can see that an explanation of $x$ is incorrect as some features with values less than 8 have positive contributions, e.g. features $x_4$ and $x_5$ have green bars, so they contribute toward the wrong class $C_1$. In our experimental evaluation we sampled instances around the corner (8,8) with $p_i \in S_0$, $i \in [1, 3]$. So, all these samples belong to the class $C_0$, hence, all features should have negative coefficients the liner model $g$. We found that LIME explanations have 25% of features with positive coefficients in $g$ that contribute to the wrong class $C_1$ which is an incorrect behaviour.* □

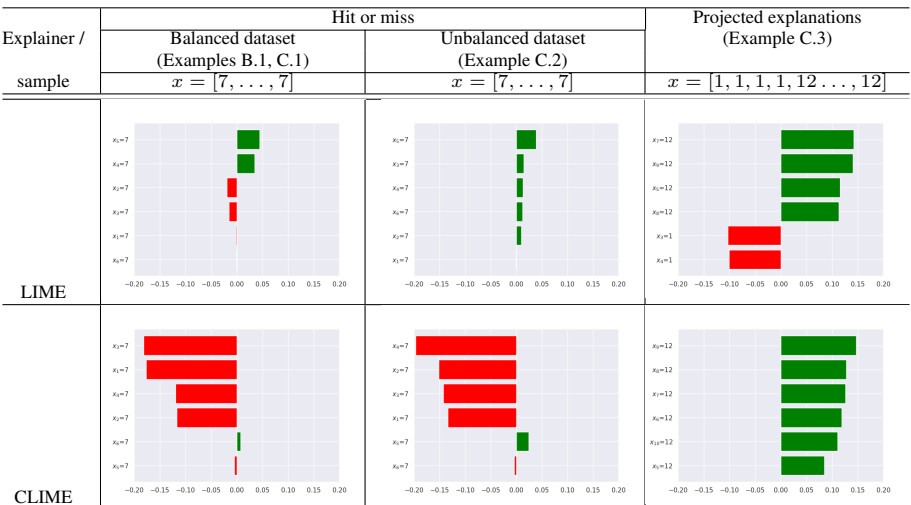

Table 1: LIME and CLIME explanations Examples B.1, C.1, C.2 and C.3.

# C  GENERATING CONSTRAINT-DRIVEN EXPLANATIONS (ADDITIONAL MATERIALS)

In this section we consider several examples to demonstrate undesired consequences of OOD sampling for LIME and how these are resolved by CLIME. Each of these examples is designed to model an abstraction of the real user-case. Moreover, for these simplified tasks, we can estimate quality of the sampling procedure using our metric $SQ$ as we can compute $W_{\mathcal{D}}$ analytically.

**Example C.1** ('Hit and miss continued'). *We come back to Example B.1 to evaluate performance of CLIME. First, we estimate our metric $SQ$: $SQ(CLIME) = 0.999$. As it is close to 1, we conclude that CLIME samples from the true distribution. Then we generated explanations produced by CLIME for the same sample $x = [(7,7),(7,7),(7,7)]$. The second row (Table 1, 'Balanced dataset' column) shows explanations produced by CLIME. Note that CLIME produces valid explanations – all features with non-negligible coefficients have negative contributions (the corresponding horizontal bars are red). The same holds for all samples $x = [p_1, p_2, p_3]$ with $p_i \in S_0$ that are in near the corner (8,8). We recall that these samples were problematic for LIME to produce correct explanations.* □

**Example C.2** ('Hit and miss'. Sensitivity to the training set). *As we described in Example B.1, LIME is sensitive to OOD samples even if we have a near to perfect classifier $f$. Moreover, we observed that the choice of the dataset can severely impact explanations. We created a new dataset for the same classification task as in Example B.1 (Figure 1). The only difference is that we have twice more $C_0$ samples compared to $C_1$ samples in the training set. Unbalanced datasets are often occur in practice if samples for one class are much cheaper to obtained compared to the other class. We compute $SQ$: $SQ(LIME) = 0.43$, which is much less than 1, and $SQ(CLIME) = 0.998$. Table 1 ('Unbalanced dataset') shows our results for $x$ in the second column . Based on our metrics, it is expected that CLIME produces better explanations as almost all features contribute to $C_0$ which is not the case for LIME.* □

**Example C.3** ('Projected explanations'). *Let us consider a scenario where the user asks for an explanation constrained on a subspace of the original distribution $\mathcal{D}$. For example, for the recidivism dataset, a user might want to know how decisions are made for the subgroup of people who are junkie or have problems with alcohol. Here, we craft a synthetic example to show how projecting on such subsets of features can affect explanations.*

***Definition of a task.*** *We consider two areas $S_0$ and $S_1$ on the 2D plane (see Figure 5). A sample $x$ from $\mathcal{D}$ represents five 2D integer points from $S_0 \cup S_1$. Hence, we have ten features vector $x = [p_1, \ldots, p_5] = [(x_1, x_2), \ldots, (x_9, x_{10})]$, where $(x_{2i-1}, x_{2i})$, $i = 1, \ldots, 5$ is a 2D point. Given a sample $x = [p_1, \ldots, p_5]$, we say that $x$ belongs to $C_1$ iff there exist $i, j \in [1, 5]$ s.t. $p_i, p_j \in S_1$ and to $C_0$ otherwise. So, if two out of five points in $x$ are in $S_1$ then $x$ is classified as 1. As in Example B.1 points in the area $S_0$ (resp. $S_1$) contribute to the $C_0$ (resp. $C_1$) class, respectively.*

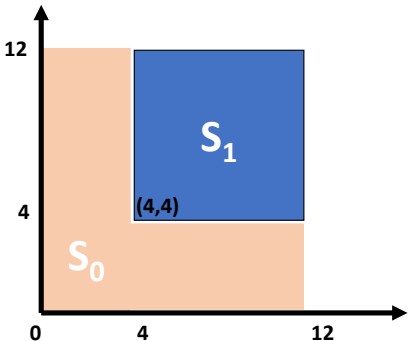

Figure 5: Projected explanations example.

***Projection constraints.*** *Next, we assume that the user adds constraints that the first two points must be in $S_0$: $p_1, p_2 \in S_0$. As we have a constrained distribution, namely, $p_0, p_1 \in S_0$, we know that the first four features, $x_1, x_2, x_3$, and $x_4$, should not appear in any explanation as no matter how we vary these features values in the constrained space, the corresponding points $p_1, p_2$ are always in $S_0$. Hence, **the corresponding coefficients of features** $x_1, \ldots, x_4$**, should be close to zero** as changing these features in the constrained space does not have an effect on the prediction.*

***Explainers performance.*** *We train an almost perfect neural network-based classifier for this simple task . We consider explanations of length six. We compute our sampling quality metric $SQ$. We get that $SQ(LIME) = 1.38$ and $SQ(CLIME) = 0.97$. Hence, these estimates signal that LIME's sampling procedure does not sample from the true distribution.*

*Table 1 shows results for LIME and CLIME in the third column ('Projected explanations') for another sample $x = [(1,1), (1,1), (12, 12), (12, 12), (12, 12)]$. As you can see, $x_3$ and $x_4$ occur in the explanation for LIME which is incorrect. CLIME produces correct explanations as constrained features $x_1, x_2, x_3$, and $x_4$ do not occur in CLIME's explanations. (See more in Section E.1).* $\square$

# D  CERTIFYING CONSTRAINT-DRIVEN EXPLANATIONS (ADDITIONAL MATERIALS)

## D.1  AA' ALGORITHM

For completeness, we present the $AA'$ algorithm in full, which is a simple adaptation of $AA$ algorithm by (Dagum et al., 2000) which uses almost-uniform samples instead of perfectly uniform. $AA'$, takes as input 3 parameters, $\varepsilon_1, \varepsilon_2$ and $\delta$. It uses $\varepsilon_2$ as the tolerance parameter in calls to an almost-uniform sampler (encapsulated in procedure $getSamples$). For ease of exposition, we use $\phi$ to denote the subspace that $getSamples$ generates samples from. We emphasize that the estimation algorithm is not limited to Boolean domains; it is applicable to any population so long as it is possible to sample (almost) uniformly from that population. As before, we assume access to a procedure $checkProperty$ which returns 1 if the property of interest is satisfied by the sample and 0 otherwise.

$$\Pr[\frac{(1 - \varepsilon_1)}{1 + \varepsilon_2}\rho \leq \widehat{\rho} \leq (1 + \varepsilon_1)(1 + \varepsilon_2)\rho] \geq (1 - \delta) \tag{9}$$

The guarantees provided by $AA'$ are similar to Eqn. 5 and are precisely captured in Eqn. 9. See Lemmas 1 and 4 for the proof.

## D.2  PROOF OF THEOREM 1

**Mean deviation due to almost uniform sampling**

**Lemma 1.** *Let $\rho$ be the density of instances that satisfy some property $P$ in a universe $\mathcal{U}^Z$, that is,*

$$\rho = \frac{\sum_{z \in \mathcal{U}^z} \mathcal{I}[P(z)]}{|\mathcal{U}^Z|}$$

**Algorithm 4** $AA'(\varepsilon_1, \varepsilon_2, \delta)$

**Output:** $\widehat{\rho}$: Estimate of $\rho$ satisfying Eqn. 9
1: $\tau \leftarrow 4(e-2)\ln(2/\delta)/\varepsilon_1^2$
2: $\tau_2 \leftarrow 1.1\tau$
3: $\mathring{\rho} \leftarrow stoppingRule(1/2, \varepsilon_2, \delta/3, \tau)$
4: $N \leftarrow \tau_2 \cdot \varepsilon_1/\mathring{\rho}$
5: $a \leftarrow 0$
6: **for** $i \in \{1, \ldots, N\}$ **do**
7: $\quad s_1 \leftarrow getSamples(\phi, \varepsilon_2, 1)$
8: $\quad c_1 \leftarrow checkProperty(s_1)$
9: $\quad s_2 \leftarrow getSamples(\phi, \varepsilon_2, 1)$
10: $\quad c_2 \leftarrow checkProperty(s_2)$
11: $\quad a \leftarrow a + (c_1 - c_2)^2/2$
12: **end for**
13: $\xi \leftarrow max(a/N, \mathring{\rho} \cdot \varepsilon_1)$
14: $N \leftarrow \tau_2 \cdot \xi/\mathring{\rho}^2$
15: $a \leftarrow 0$
16: **for** $i \in \{1, \ldots, N\}$ **do**
17: $\quad s \leftarrow getSamples(\phi, \varepsilon_2, 1)$
18: $\quad c \leftarrow checkProperty(s)$
19: $\quad a \leftarrow a + c$
20: **end for**
21: $\widehat{\rho} \leftarrow a/N$
22: **return** $\widehat{\rho}$

**Algorithm 5** $stoppingRule(\varepsilon_1, \varepsilon_2, \delta, \tau)$

**Output:** $\mathring{\rho}$ (weak estimate)
1: $\tau_1 \leftarrow 1 + (1+\varepsilon_1)\tau$
2: $N \leftarrow 0$
3: $a \leftarrow 0$
4: **while** $a < \tau_1$ **do**
5: $\quad s \leftarrow getSamples(\phi, \varepsilon_2, 1)$
6: $\quad a \leftarrow a + checkProperty(s)$
7: **end while**
8: $\mathring{\rho} \leftarrow \tau_1/N$
9: **return** $\mathring{\rho}$

*Suppose we sample each instance $z \in \mathcal{U}^Z$ almost-uniformly, that is*

$$\frac{1}{(1+\varepsilon)|\mathcal{U}^Z|} \leq Pr[z^* = z] \leq \frac{1+\varepsilon}{|\mathcal{U}^Z|}$$

*Then we have*

$$\frac{\rho}{(1+\varepsilon)} \leq \sum_{z \in \mathcal{U}^Z} \mathcal{I}[P(z)] \cdot \Pr[z^* = z] \leq (1+\varepsilon) \cdot \rho$$

*Proof.* In the worst cases, each sample $z$ s.t. $\mathcal{I}[P(z)] = 1$ will be sampled with

1. probability $\frac{1+\varepsilon}{|\mathcal{U}^Z|}$, in which case $\sum_{z \in \mathcal{U}^Z} \mathcal{I}[P(z)] \cdot \Pr[z^* = z] = (1+\varepsilon) \cdot \rho$

2. probability $\frac{1}{(1+\varepsilon)|\mathcal{U}^Z|}$ in which case $\sum_{z \in \mathcal{U}^Z} \mathcal{I}[P(z)] \cdot \Pr[z^* = z] = \frac{\rho}{(1+\varepsilon)}$

$\square$

**Lemma 2.** *Let $\rho \leq \gamma - \varepsilon$. Then the probability that $checkThreshold$ returns True is at-least $1 - \delta$.*

*Proof.* Note that $\mu_C \leq (1 + \varepsilon/2)\rho$ as $getSamples$ may return almost uniformly distributed samples in line 3 of $checkThreshold$. We will first prove that $\mu_C + \nu \leq \gamma$. We have

$$\gamma \leq \gamma$$
$$\implies \gamma(1 + \varepsilon/2 - \varepsilon/2) \leq \gamma$$
$$\implies \gamma(1 + \varepsilon/2) - \gamma\varepsilon/2 \leq \gamma$$

But we have $\gamma \geq \rho + \varepsilon$. Therefore,

$$(\rho + \varepsilon)(1 + \varepsilon/2) - \gamma\varepsilon/2 \leq \gamma$$
$$\implies \rho + \rho\varepsilon/2 + \varepsilon + \varepsilon^2/2 - \gamma\varepsilon/2 \leq \gamma$$
$$\implies \mu_C + \nu \leq \gamma$$

The last equation follows from the fact that $\mu_C \leq (1+\varepsilon/2)\rho$ and $\nu \leq \varepsilon + \varepsilon^2/2 - \gamma\varepsilon/2$ from line 1 of $checkThreshold$. Now since $\mu_C + \nu \leq \gamma$, we have $\Pr[C \leq \gamma] \geq \Pr[C \leq \mu_C + \nu] = 1 - \Pr[C \geq \mu_C + \nu]$. $C$ is the average of $N$ independent 0/1 random variables $c$ (line 6). Therefore applying Chernoff bound,

$$\Pr[C \geq \mu_C + \nu] \leq exp\{-2\nu^2 N\} \tag{10}$$

But $N = \frac{1}{2\nu^2} \log(\frac{1}{\delta})$. Therefore, $\Pr[C \geq \mu_C + \nu] \leq \delta$. Substituting back, we get $\Pr[C \leq \gamma] \geq 1-\delta$. Therefore, in line 8, the probability that $checkThreshold$ returns True, is at least $1 - \delta$.

$\square$

**Lemma 3.** *Let $\rho \geq \gamma + \varepsilon$. Then the probability that $checkThreshold$ returns False is at-least $1 - \delta$.*

*Proof.* Similar to preceding Lemma. $\square$

**Lemma 4.** *Algorithm $AA'$ returns an estimate $\widehat{\rho}$ such that $\Pr[(1-\varepsilon)\rho_\psi \leq \widehat{\rho}_\psi \leq (1+\varepsilon)\rho_\psi] \geq (1-\delta)$*

*Proof.* Algorithm $AA'$ takes as input 3 parameters, $\varepsilon_1$, $\varepsilon_2$ and $\delta$. It invokes the $AA$ Algorithm by Dagum et al. with parameters $\varepsilon_1$ and $\delta$ on samples generated by an almost-uniform sampler with parameter $\varepsilon_2$. By Lemma 1, the population mean can shift at most by a factor of $(1 + \varepsilon_2)$ due to almost-uniform sampling (instead of perfectly uniform). Combined with the approximation guarantees of $AA$ algorithm, the resulting tolerance has an upper-bound of $(1 + \varepsilon_1) \times (1 + \varepsilon_2)$ and a lower bound of $(1 - \varepsilon_1)/(1 + \varepsilon_2)$. In line 4 of Alg. 2, the $AA'$ algorithm is invoked with $\varepsilon_1 = \varepsilon_2 = 0.4 * \varepsilon$. Substituting these values in the expressions for the upper and lower bounds on tolerance, we get the result. $\square$

**Theorem 1.** *If $\rho \leq \gamma - \varepsilon$, then $computeEstimate$ returns $\perp$ with high probability (i.e. at least $1 - \delta$). If $\rho \geq \gamma + \varepsilon$, w.h.p., it returns an estimate $\widehat{\rho}$ such that $\Pr[(1-\varepsilon)\rho \leq \widehat{\rho} \leq (1+\varepsilon)\rho] \geq (1-\delta)$.*

*Proof.* Follows from preceding lemmas. $\square$

Note that the bounds and number of samples used for proving the preceding theorem were computed assuming we only have access to Almost-Uniform samples. The bounds can be made significantly tighter or the number of samples can be reduced, if we have access to perfectly uniform samples.

### D.3 Applicability of the Estimation framework

We emphasize that our estimation framework is general enough to compute any metric for any universe (so long as one can sample from it almost uniformly) according to the guarantees provided by Thm. 1 in any setting (not just explainability). Further, for the specific application explainability, our estimation framework can be used for measuring properties like fidelity of any explainer model (not just the ones crafted by CLIME), and on any subspace of inputs (not just the one that the explainer was trained on). For example, the fidelity of a CLIME explainer model trained on a subspace defined by one set of constraints (say $\phi$) maybe evaluated on another subspace defined by $\psi$. If the fidelity on $\psi$ is found to be high enough, it can save the cost of having to generate a separate explanation for $\psi$. This can be especially useful in model debugging where users may refine constraints frequently.

### D.4 Empirical Evaluation of Estimation Algorithm

In order to test the scalability of our estimation algorithm (Algs. 2, we evaluated its performance on the same set of benchmarks used by (Narodytska et al., 2019).

**Benchmarks** The benchmarks are CNF formulas that encode fidelity of Anchor (Ribeiro et al., 2018) explanations for Binarized Neural Networks with upto 3 hidden layers and 100 neurons in total, and are generated from Adult, Recidivism and Lending datasets. There were 50 CNF formulas from each dataset, for a total of 150 benchmarks, with number of variables ranging between 20,000 to 80,000 and number of clauses ranging between 80,000 and 290,000. The projected model count of each formula represents the number of inputs on which the class-label for Anchor's explanation matches the true label of the instance being explained. The fidelity of an explanation can thus be computed as the ratio of the solution-count of the formula, to the size of the universe.

**Parameters** We used the same tolerance ($\varepsilon = 0.8$) and confidence ($\delta = 0.2$) used in (Narodytska et al., 2019), for the main experiment. Additionally, we set the threshold to $\gamma = 0.05$. We also compared the running times for tighter tolerance and confidence (see Discussion below).

**Experimental Setup** We set a time out of 3 minutes (180 seconds), and ran each experiment on Intel Xeon E5-2650 CPU running at 2.20GHz, with 4GB main memory. We compiled our code using GCC 6.4 with O3 flag enabled. For ApproxMC we used the latest publicly available version (4.01).

**Results** The results are presented in Fig. 2. Each point in blue corresponds to one benchmark, and the x-coordinate represents the time taken by ApproxMC while the y-coordinate represents the time taken by our approach. It can be seen that our approach completes all benchmarks in under 25 seconds with majority taking less than 10 seconds. In contrast, ApproxMC is able to finish only 10 benchmarks out of 150 in under 25 seconds, with a majority taking around 75 seconds. The average (geometric mean) speedup factor offered by our tool relative to ApproxMC was 7.5.

**Discussion** Fig. 2 conclusively demonstrates the efficiency of our approach as compared to hashing and SAT based approaches like ApproxMC. Our tool was able to return estimates with two-sided bounds for all benchmarks. Our tool offers an average (geometric mean) speedup by a factor of 7.5 relative to ApproxMC. In addition, our approach is able to scale far better than ApproxMC for tighter tolerance and confidence parameters. For a representative benchmark, we evaluated the time taken by our tool and ApproxMC for the default tolerance ($\varepsilon = 0.8$) and confidence ($\delta = 0.2$). We then computed the slow-down in both tools after setting $\varepsilon = 0.05$ and confidence $\delta = 0.1$. We found that our tool slowed down by a factor of 10 while ApproxMC slowed down by a factor of 30. Lastly, we computed the error in the estimate returned by our algorithm (with default tolerance and confidence) on small benchmarks where it was possible to compute the true count. We consistently observed that the error was less than 0.1, which is much smaller than the specified tolerance of 0.8. Thus our approach is both sound and efficient in practice.

## E EXPERIMENTS(ADDITIONAL EXPERIMENTS)

We perform an extensive evaluation of our explainer CLIME. First, we consider explanations for a classification task using synthetic and real-world datasets. Second, we demonstrate how our framework can be used to detect adversarial attacks on an explainer.

### E.1 SYNTHETIC EXPERIMENTS

For each synthetic dataset, we generated train and test datasets with 1000 samples per class. We trained a MLP with three internal layers of fifty neurons that gives 99.9% on the test set. So, we have an almost perfect classifier.

#### E.1.1 'HIT OR MISS' (EXAMPLE B.1)

We consider a binary classification task from Example B.1. We recall that a sample $x$, $x \sim \mathcal{D}$, represents integer coordinates of three points $p_1, p_2$ and $p_3$ drawn from $S_0 \cup S_1$. Hence, we have six features vector $x = [x_1, x_2, \ldots, x_6]$, where $p_i = (x_{2i-1}, x_{2i})$, $i = 1, 2, 3$, is a 2D point.

*Computation of SQ.* As we know the true distribution, we can compute $W_{\mathcal{D}}$. Let $hit(x) = |\{p_i | p_i \in S_1, p_i \in x\}|$ be a function that computes the number of points $p_i \in x \cap S_1$. For example, $hit([(1,1), (10,10), (10,9)]) = 2$ as $(10,10) \in S_1$ and $(10,9) \in S_1$. We have that $P(p_i \in S_1) = \frac{|S_1|}{|S_0|+|S_1|} = 25/(25+49) = 0.34$. Hence,

$$W_{\mathcal{D}} = 3P(hit(x) = 1) + 3P(hit(x) = 2) + P(hit(x) = 3) = 3 \times 0.15 + 3 \times 0.07 + 0.04 = 0.7.$$

We generate 100 samples such that $x_i \in [5, 7]$, $i = [1, 6]$. Therefore, each sample consists of points that are close to the corner (8,8) and it is classified as $C_0$. We compute explanations by LIME and CLIME on these samples. We empirically compute $W_{\mathcal{Z}'}^{LIME}$ and $W_{\mathcal{Z}'}^{CLIME}$ for the neighborhood $\mathcal{Z}'$. We found that $W_{\mathcal{Z}'}^{LIME} \approx 0.53$ and $W_{\mathcal{Z}'}^{CLIME} \approx 0.71$ So, $\widehat{SQ}(LIME) \approx$

$0.75 < 1$ and $SQ(CLIME) \approx 1$. These results hint that LIME may produce invalid explanations. We experimentally verify this hypothesis next.

*Explainers performance.* We recall that each sample consist of points that are close to the corner (8,8) and it is classified as $C_0$. Hence, we expect that coefficients in the explanations are less than 0 as all features contribute to $C_0$. To evaluate this expected behavior, we computed

$$C^+ = \frac{\sum_{i=1,c_i>0}^{6} c_i}{\sum_{i=1}^{6} |c_i|},$$

where $c_i$ is the $i$th normalised (between 0 and 1) coefficient in an explanation (Eqn. 2). We average results over 100 samples. Ideally, $C^+$ has to be equal to zero, as we *should not have positive coefficients* in an explanation in any of these samples. We get that $C^+_{LIME} = 0.18$ and $C^+_{CLIME} = 0.01$. As you can see, $C^+_{LIME}$ is significantly greater than zero showing that many explanations are incorrect. Therefore, explanations produced by LIME suffer from ODD sampling and result in incorrect explanations.

### E.1.2 Projected explanations (Example C.3)

We consider a binary classification task from Example C.3. We recall that a sample $x$ from $\mathcal{D}$ represents five 2D integer points from $S_0 \cup S_1$. Hence, we have ten features vector $x = [p_1, \ldots, p_5] = [(x_1, x_2), \ldots, (x_9, x_{10})]$, where $p_i = (x_{2i-1}, x_{2i})$, $i = 1, \ldots, 5$ is a 2D point. We recall that the user constraint is that $p_1, p_2 \in S_0$.

*Computation of SQ.* We define $hit(x)$ the same way as above. We have that $P(p_i \in S_1) = \frac{|S_1|}{|S_0|+|S_1|} = 64/144 = 0.44$ for this task. Hence,

$$W_{\mathcal{D} \cap \{p_1, p_2 \in S_0\}} = 3 \times P(hit(x) = 2) + P(hit(x) = 3) = 0.33 + 0.08 = 0.41.$$

We compute explanations by LIME and CLIME on 100 samples that satisfy the user constraint. We ask explainers to give us the top six features per sample. Moreover, to help LIME to capture constraints, we provide it with a dataset where $p_1, p_2 \in S_0$ for all samples. Then we run LIME and CLIME sampling procedures and compute $W_{\mathcal{Z}'}^{LIME}$ and $W_{\mathcal{Z}'}^{CLIME}$ for the neighborhood $\mathcal{Z}'$. We found that $W_{\mathcal{Z}'}^{LIME} \approx 0.56$ and $W_{\mathcal{Z}'}^{CLIME} \approx 0.40$ So, $SQ(LIME) \approx 1.38 > 1$ and $SQ(CLIME) \approx 1$. Again, we can see that $SQ(LIME)$ is significantly differ from 1 indicating ODD sampling.

*Explainers performance.* We run LIME and CLIME on 100 samples and compute the number of times each feature occurs in an explanation (an explanation contains 6 features). Figure 6 shows our results. For each feature, we show the number of occurrences of this feature in produced explanations. Namely, the blue histogram shows results for LIME and the orange histogram shows results for CLIME. As we discussed before, as $p_1$ and $p_2$ are in $S_0$, so the first 4 features should not appear in the explanation as changing their values never affects the prediction. Indeed, CLIME almost never choose features $x_1, \ldots, x_4$ in the explanation (almost zero height orange bars for these features). However, LIME often picks these constrained features in its explanations.

### E.2 Adult dataset

The *Adult* dataset (Kohavi, 1996) is originally taken from the Census bureau and targets predicting whether or not a given adult person earns more than $50K a year depending on various attributes, e.g. race, sex, education, hours of work, etc. We pre-processed columns with continuous features, e.g. the pre-processor discretizes the capital gain and capital loss features into categorical features, e.g. 'Unknown', 'Low' and 'High'(Ribeiro et al., 2018). We trained a MLP with three internal layers of 64, 32, and 32 neurons that achieves 84.2% on the test set. We consider the two constrained spaces:

- 'female space' := (CAPITAL LOSS = 'UNKNOWN') AND (CAPITAL GAIN = 'UNKNOWN') AND (EDUCATION = 10TH OR 11TH GRADE ) AND (SEX = FEMALE)

- 'male space' := (CAPITAL LOSS = 'UNKNOWN') AND (CAPITAL GAIN = 'UNKNOWN') AND (EDUCATION = 10TH OR 11TH GRADE ) AND (SEX = MALE).

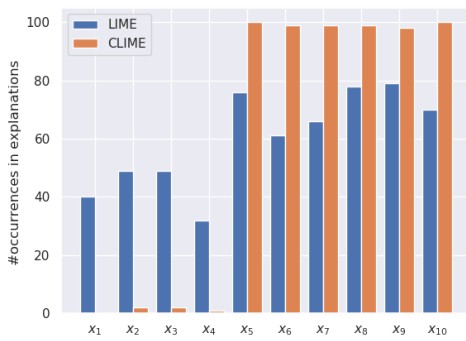

Figure 6: Occurrences of features in explanations (Example C.3).

We compute explanations for 100 samples within the corresponding constrained space. Figure 7 shows results for LIME (a) and CLIME (b), respectively, for both constrained spaces. At each plot, the blue histogram shows results for 'female space' and the orange histogram shows results for 'male space'. Each bar shows the average rank of a feature in explanations, e.g. if the height of the bar is 0.1 than the corresponding feature is mostly top ranked in all explanations. We used the following shortcuts of features names in Figure 7 : 'Age' ('Age'), 'Workclass' ('WrkC'), 'Education' ('Edu'), 'Marital Status' ('Mard'), 'Occupation' ('Occp'), 'Relationship' ('Relp'), 'Race' ('Race'), 'Sex' ('Sex'), 'Capital Gain' ('CapG'), 'Capital Loss' ('CapL'), 'Hours/week' ('Hrs'), and 'Country' ('Cntr'). First, we observe that as features Capital Loss, Capital Gain and Sex are fixed in both constrained sub-spaces so their values should not influence the outcome. Note that these features are always ranked among last by CLIME. However, this is not the case for LIME. We observed exactly the same behavior in our synthetic example from Example C.3. Second, we can see that LIME explanations are not so much different for both spaces. While we do not have the ground truth to check against in this example, we can contrast LIME results with CLIME results. CLIME shows that 'Education' is more important for males than for females (see in Figure 7(b), the orange bar is lower than the blue bar for the 'Education' feature). 'Relationship' seems to be more important for females than for males (see in Figure 7(b), the orange bar is higher than the blue bar for the 'Relationship' feature). These results indicate that there might be a potential bias in these spaces.

### E.3 DETECTING ADVERSARIAL ATTACKS(EXTENDED)

Suppose a biased classifier makes decisions solely based on some sensitive feature; for ex: a classifier for accepting / rejecting loan applications that decides purely based on an individual's race. LIME's explanations for the predictions made by such a classifier would clearly indicate race to be a major deciding factor. (Slack et al., 2020) devised a technique that takes such an arbitrary biased classifier and constructs another classifier that closely mimics the biased classifier on in-distribution data, yet is able to evade bias-detection by preventing the sensitive feature from appearing in post-hoc explanations. Note that accurately measuring the fidelity of the explainer model alone is not sufficient to detect such an attack as the problem lies in the way the opaque classifier is constructed and not in the resulting explanation. We first discuss the construction of the adversarial classifier, and then see how constraints can help in foiling it.

### E.3.1 BACKGROUND

The attack makes use of the fact that LIME's perturbation procedure creates samples that are distinguishable from those of the original distribution. The first step is to train a classifier such as a Random Forest to detect whether a given sample is in or out of distribution. This classifier is trained on a synthetic data-set consisting of instances from the original data-set labeled as in-distribution and LIMEs perturbations of each of those instances as OOD. This OOD detecting classifier is able to achieve very good accuracy, which is a testament to the severity of the OOD problem of LIME.

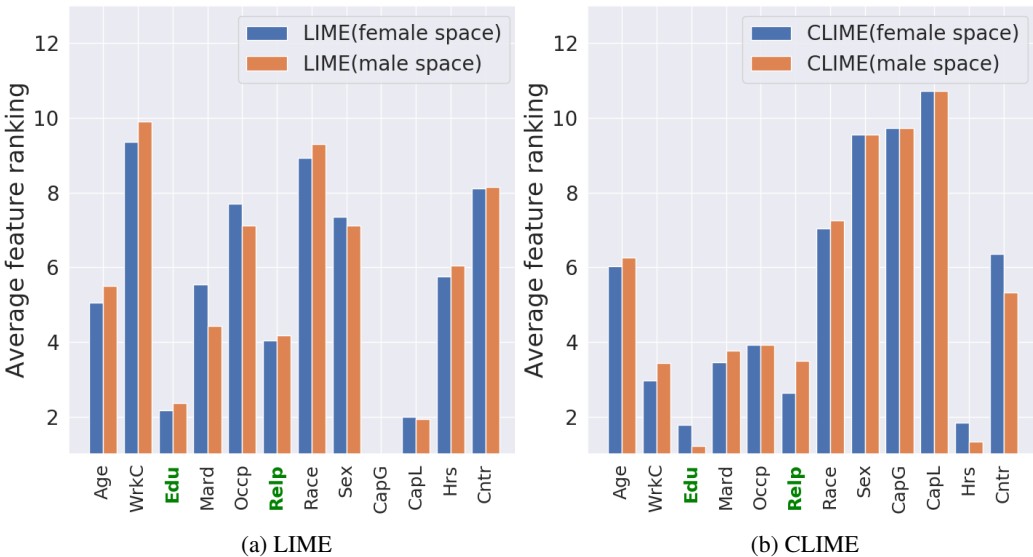

(a) LIME          (b) CLIME

Figure 7: Explanations for the adult dataset on female and male spaces.

The next step in the adversarial attack is to create an innocuous classifier that makes decisions based on some unrelated feature. For example, in case of credit card applications, the innocuous classifier could make decisions based on either an existing unimportant feature such as card number or on a new feature added to the dataset whose values are randomly sampled. This unrelated feature will be made to appear in place of the sensitive feature in explanations.

The last step of the attack is construct the adversarial classifier which first applies the OOD-detecting classifier on a given instance $x$. If the OOD-detecting classifier predicts $x$ to be in-distribution, the the adversarial classifier outputs the prediction of the biased classifier on $x$. If $x$ is predicted to be OOD, then the adversarial classifier outputs the prediction of the innocuous classifier on $x$.

Clearly, the predictions made by the adversarial classifier on in-distribution instances are biased, as the adversarial classifier simply outputs the biased classifiers prediction. Suppose LIME is used for explaining the prediction made by the adversarial classifier on some in-distribution instance $x$. LIME first perturbs $x$. However, due to LIME's OOD sampling problem, most of the perturbed samples are OOD and are detected by the OOD-detecting classifier as such. Therefore, the output of the adversarial model on the perturbed samples is that of the innocuous classifier. Thus for vast majority of the samples on which LIME's explainer model is trained, the unrelated feature appears as the deciding factor for the adversarial model's prediction, whereas the sensitive feature is successfully hidden.

Experimentally, the authors generated explanations for the adversarial model on multiple test instances belonging to Recidivism, Communities and Crime and German datasets. They showed that the sensitive features appeared with very low frequency among the top 3 explanations for all the test points, which indicates that the attack was successful.

### E.3.2   EXPERIMENTAL SETUP

We hypothesized that for discrete data, instances with low Hamming distance to points in the given dataset should tend to be in-distribution, whereas instances with a high hamming distance should tend to be OOD. Note that the recommended way for using LIME is on data which is discretized. LIME itself uses Hamming distance as a distance metric for weighting the samples it generates. However, it is unable to specifically sample proportionately from high-weight regions which leads to a lot of samples with low weights. With CLIME one can, however, use constraints to generate samples that would be weighted highly by LIME. For a given Hamming distance $h$ and a set of instances $S$, a

formula $\phi^h$ can be constructed such that the the solutions of $\phi^h$ are exactly those instances that are at most at a distance $h$ from each instance in $S$. If our hypothesis is correct, the sensitive features should appear as the top CLIME explanation of the adversarial classifier for low values of $h$, while innocuous features should dominate for larger $h$.

Note that the Hamming distance between two instances doesn't take into account the degree to which the value of a feature in one instance differs from the other. Specifically, for discrete features that are the quantized versions of continuous features, values of adjacent quantiles are clearly 'closer' that those of non-adjacent quantiles. This is also true for natively discrete features like the star rating of a product. One can design metrics smarter than the hamming distance, that also take into account these differences in values and are even more robust against adversarial attacks. In this work however, we focus on the Hamming distance as a proof of concept and leave experimentation with other metrics to future work.

### E.4 PRACTICAL CONSIDERATIONS

The tool by (Slack et al., 2020), as of the time of writing this paper, was only able to create an adversarial attack on continuous features (for LIME), and did not have the ability to train the OOD-detecting classifier on discrete data. For our experiments, we augmented their tool to handle purely discrete data by using LIME's native perturbation procedure for generating training data for the OOD-detecting classifier. For discrete data, the parameter 'perturbation_multiplier' in their tool plays an important role. Its default value of 30 creates a very imbalanced dataset and the accuracy of the OOD-detector is dramatically affected and it classifies most instances as OOD. We found that a value of 2 led to the best test accuracy for the OOD-detector.

With discrete data, we also observed that the efficacy of the adversarial attack on LIME was drastically reduced. The sensitive feature appeared as the 2nd from top explanation almost 100% of the time. Additionally, for attacks that used 2 uncorrelated features, the sensitive feature also appeared as the top explanation with a considerable frequency. However, for attacks that used 1 uncorrelated feature, the top explanation was the uncorrelated feature with almost 100% frequency. Hence in our experiments, we focus on the top explanation ((Slack et al., 2020) considered top 3 explanations) and use only one unrelated feature. For CC and German experiments, we selected the top 12 most important features of the 3 datasets using scikit-learn's Random Forest classifier, as there were far too many features in the original datasets (100 for CC and 28 for German). The test accuracy of the Random Forest with 100 estimators was 0.83 and 0.75 for CC and German respectively even after restriction to 12 features, down from 0.875 and 0.82, respectively, before removing features.

### E.5 RESULTS

The results for the Recidivism dataset were depicted in Figure 4. The results for CC dataset are depicted in Fig. 8 and those for German in Fig. 9. It can be seen that the sensitive feature appears prominently for low hamming distance while the unrelated feature dominates higher for higher distance, which confirms the efficacy of CLIME.

In general, it is harder to craft attacks similar to (Slack et al., 2020) against CLIME explanations, as the sampling procedure is dependent on user-provided constraints and is not fixed like LIME. However, for the current experiment, an adversary could conceivably train the OOD-detector on instances with low hamming distance from the original dataset to specifically foil our method. We tested whether the sensitive feature still appeared in the top CLIME explanation when the OOD-detector was trained in this manner (instead of being trained on LIME generated perturbations). We compare the frequency of seeing the sensitive feature as the top CLIME explanation of the adversarial classifier in the two scenarios in Table 2. While the frequency of seeing the sensitive feature drops as compared to the case when the OOD detector is trained on LIME perturbations, the drop is not very steep and the sensitive feature still figures prominently as the top explanation. This shows that our hamming distance based technique is not only able to detect attacks, but is also robust against attacks itself.

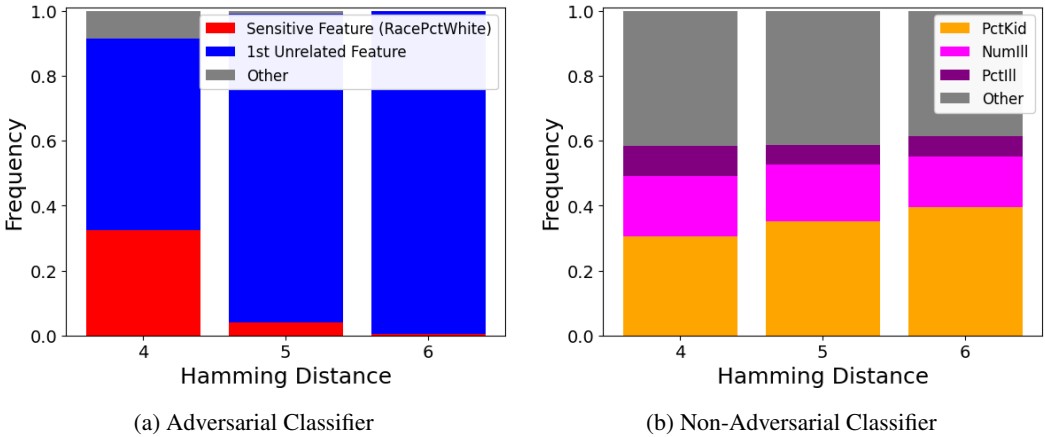

(a) Adversarial Classifier

(b) Non-Adversarial Classifier

Figure 8: CC Dataset: Top CLIME explanation vs. Hamming Distance

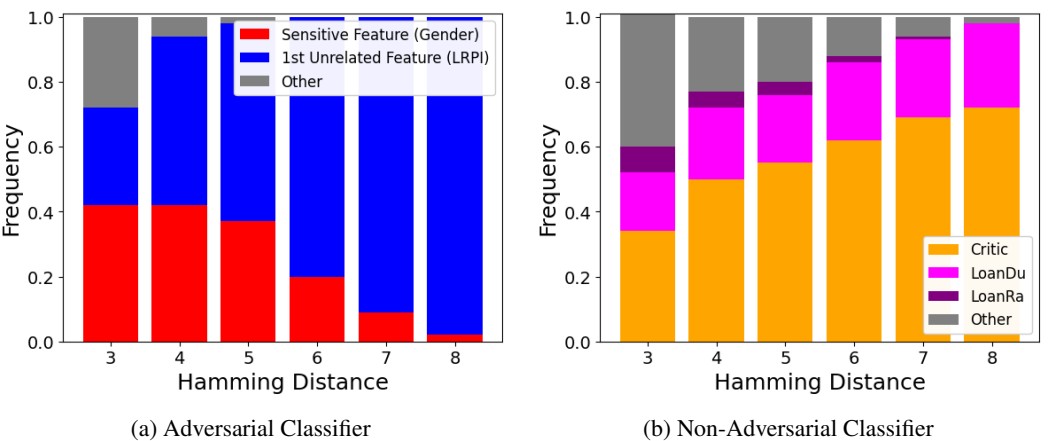

(a) Adversarial Classifier

(b) Non-Adversarial Classifier

Figure 9: German Dataset: Top CLIME explanation vs. Hamming Distance

|  | Recidivism | CC | German |
|---|---|---|---|
| OOD detector trained on LIME's samples | 0.635 | 0.32 | 0.42 |
| OOD detector trained on samples from Hamming-constraints | 0.63 | 0.1 | 0.42 |

Table 2: Frequency of sensitive feature in top explanation

