# OpenReview forum: "Constraint-Driven Explanations of Black-Box ML Models"
_ICLR.cc/2021/Conference — Reject_

### Official Review · AnonReviewer1 · 2020-10-26
**proposes simple metrics and solutions for defining locality in LIME, but lacking in clarity in examples and in relation to previous works**

**Rating:** 5
**Confidence:** 4

**Review:**

Summary: This paper proposes a new sampling method for LIME based on user-defined Boolean subspaces. They show that using these subspaces rather than the default sampling settings of LIME can lead to robustness against adversarial attacks and allow users to better undercover bugs and biases in subspaces relevant to the user's task. They additionally propose a new metric for measuring the quality of an explanation.

Pros: The authors create a novel link between the explainability literature and Boolean Satisfiability. The proposed methodology is shown to be relevant in several different applications. They propose simple metrics for evaluating the quality of samples and the quality of an explanation.

Cons: I don't have a lot of experience with ICLR, but it is not obvious to me that this paper is appropriate for this venue.

None of the metrics used in the paper seem to be weighted by density/distance to the point being explained, as is done in LIME. Given that the point of LIME is that the function is unlikely to be globally approximable by a linear function, the lack of incorporation of a weighting function seems to make this framework inferior to that of LIME (and in fact, theoretically, the binary subspaces used in this paper are merely a specific instantiation of the flexible weighting function used in LIME, the paper, as opposed to LIME, the software package). I would expect that without such a weighting function, in many cases explanations with similar values of the rho metric may vary widely in their usefulness as local explanations.

The paper suffers somewhat from relegating all of the examples to the supplementary material. Examples which are important to the points being made should be brought back into the main body of the paper.

Certain relevant works seems to have been missed: Sokol et al. (https://arxiv.org/pdf/1910.13016.pdf) proposed user-specified local surrogate explainers, including allowing users to define their own sampling subspace (but do not propose algorithms for sampling). Zhang et al. (https://arxiv.org/pdf/1904.12991.pdf) show that LIME does not accurately reflect local invariance to globally important variables.

In the first experiment, without some ground truth knowledge as to what the classifier is doing, it is not obviously useful to point out that the CLIME identifies race as a top feature for females in the recidivism dataset. The setup of Zhang et al. may be preferred, where the ground truth behavior of the classifier is known.

In the adversarial attack experiment, it is not completely clear what is done: did you use CLIME to generate explanations of the adversarial classifier from Slack et al.? Was the adversarial classifier trained with access to CLIME perturbation functions, or was it trained assuming LIME perturbation functions? This doesn't seem to immediately show superiority over the LIME framework, as at larger Hamming distances the bias is still hidden. I think a more appropriate comparison would allow the adversarial classifier access to the relevant perturbation function and consider accuracy at a variety of neighborhood widths for LIME, as with the Hamming distance.

###post rebuttal###
I have read the updated version of the paper and still feel that this paper may have errors regarding the flexibility and purpose of LIME. The idea is nice, but the paper and evaluations would benefit from more polishing before publication. I maintain my original score.

Misrepresentation of LIME:

Section 3: LIME does not assume that the sampling neighborhood is the same as the true distribution. It may assume something weaker, such as that the function being explained is fairly smooth in the sampling neighborhood. Note that this can be a feature of LIME and not necessarily a bug: if for example x1 and x2 are fully correlated in the data distribution but the classifier only uses x1, it would be impossible to tell this if sampling only within the data distribution. By sampling outside the data distribution it becomes apparent that the classifier is using x1 only. Also, LIME assumes black box access to the function, so I don't fully understand your statement that "we generally do not have access to the true labels of instances generated through sampling". It seems like you may be defining the "correct" explanation with respect to the true data distribution, rather than to the classifier. LIME is meant to explain a black-box classifier. If the classifier is wrong, LIME should reveal what the classifier does (that is, the explanation should also be "wrong" with respect to the true data). The "framework capabilities" is also simply not true: users can define the data point to be explained, as well as their own similarity kernel and/or kernel width.

Evaluation:

It's not entirely obvious to me how we can be sure that CLIME is producing the "right" explanation in C.1, C.2 without knowing the function f. If changing the training set changes the classifier f, then it is correct that the explanation should change. As mentioned above, evaluating whether or not an explanation is "correct" should be done with respect to the classifier, not the underlying data distribution. In "Detecting Adversarial Attacks", it's not clear from the text whether or not you retrain the adversarial attack with your perturbation function. Further, I suspect that LIME may also be able to identify the sensitive feature for sufficiently small neighborhood sizes when sampling in binary space Z'. It seems like a straw man argument to compare an optimized version of your sampling procedure to the default version of lime.

Minor: Equations 1) and 2), if they are describing the usage in Ribeiro et al., should include a weighting function.

Figure 2 seems not to be explained in the text and would benefit from more description.

---

> ### Author Response · Authors · 2020-11-13
> **Distance Metric, Weighted Sampling, CLIME perturbation adversaries, Venue**
>
> Thank you for your feedback and the pointers to missing references. We will upload an updated version of the paper in few days taking into account your comments and suggestions.
>
> **Q1.1 [Distance Metric in CLIME]**
> CLIME framework supports a weighting function, as the underlying sampling tools like WAPS and WeightGen natively allow weighted CNF formulas.
> Additionally, it is possible to work exclusively in the domain of unweighted formulas[Dudek et al., Chakraborty et al.]. In this work, we found unweighted formulas to be sufficient for our experiments. We discussed the distance metric in the beginning of Sec. 4.1 and we will expand the discussion to clarify this misunderstanding.
>
> [Dudek et al.]  Taming Discrete Integration via the Boon of Dimensionality Jeffrey M. Dudek, Dror Fried, and Kuldeep S. Meel Proceedings of Neural Information Processing Systems (NeurIPS), 2020.
> [Chakraborty et al.] Supratik Chakraborty, Dror Fried, Kuldeep S. Meel, Moshe Y. Vardi. From Weighted to Unweighted Model Counting. In Proceedings of the Twenty-Fourth International Joint Conference on Artificial Intelligence, IJCAI 2015: 689-695
>
>
> **Q1.2 [Advantages of CLIME weighted sampling procedure]**
> An important subtle point about LIME's sampling procedure is that it does not sample neighboring instances in proportion to their weight specified by the weighting function. As a compromise, LIME samples neighboring instances without taking the weights into account, and reweights them later when training the explainer model. This can result disproportionately many low-weight samples, which defeats the notion of locality. In contrast, CLIME can directly perform weighted sampling. We would be happy to discuss this in more detail as needed.
>
>
> **Q2: [Clarification on experiments]**
> We had used the Appendix for a deeper discussion of the experiments due to lack of space. We will use an additional page to expand on experiments.
>
> **Q3: [Clarification whether "adversarial classifier trained with access to CLIME perturbation functions"]**
> Yes, we evaluated CLIME's robustness to the adversarial classifier trained with access to CLIME perturbation function, and compared it to the case when using using LIME's perturbation procedure. For CLIME, we used constraints enforcing lowest Hamming distance for the respective experiment (i.e. the left-most column of each graph). The results are presented in Table 3 (Appendix D.5). We observe that the sensitive feature still features prominently as the top explanation which shows that CLIME continues to be robust even in this case.
>
> Thank you for the pointers to the two papers we had missed. We will incorporate them in the final version.
>
> **Q4: [ICLR venue]**
> We would like to point out that a number of paper have been presented in ML conferences on related topic, that employ optimization/feasibility techniques.
> Please see few examples from ICLR below. Therefore, we believe our work on enhancing arguably the most popular explainer will be interesting to the ICLR audience.
> ICLR'19: Combinatorial Attacks on Binarized Neural Networks
> Elias B Khalil, Amrita Gupta, Bistra Dilkina
>
> ICLR'19: Learning a SAT Solver from Single-Bit Supervision
> Daniel Selsam, Matthew Lamm, Benedikt Bunz, Percy Liang, Leonardo de Moura, David L. Dill
>
> ICLR'20: In Search for a SAT-friendly Binarized Neural Network Architecture
> Nina Narodytska, Hongce Zhang, Aarti Gupta, Toby Walsh

---

> > ### Comment · AnonReviewer1 · 2020-11-18
> > **Response**
> >
> > Thanks for your clarifications. My remaining concerns are:
> >
> > 1) defining a subspace/neighborhood is not a unique feature of your tool. The neighborhood definition in LIME is very flexible, and if users knew they were interested in a specific subspace it would not be that difficult to define a LIME sampling procedure that fits the subspace (one could even imagine sampling and then accepting/rejecting points based on whether they fit the criteria). Results that focus on whether or not the uniform sampling within that subspace are more useful than LIME samples would be more relevant to the paper. To define 'usefulness' seems like it would require either ground truth knowledge of the function you are approximating or user studies.
> >
> > 2) again, I believe a primary issue with neighborhoods in LIME is not that they are difficult to sample from if you know how to describe them - it is more that users do not know how to describe a neighborhood that is relevant to a given task (as pointed out also by Rev2). I agree that this argument is fairly unconvincing without some user study showing that users are somehow better able to define the boolean subspace relevant to a task than the "neighborhood" as described by LIME.
> >
> > It may be useful to devote more attention to theoretical results and examples and also to more narrowly define the point of your experiments (that is, focus on superior sampling rather than subspace definition).

---

> > > ### Author Response · Authors · 2020-11-20
> > > **Rejection Sampling, Ease of Sampling, Writing Constraints in Practice**
> > >
> > > Thank you for your comments.
> > >
> > > **Q1: [Retrofitting LIME with Rejection Sampling]**
> > > Firstly, we emphasize that LIME's sampling procedure is quite rigid, and affords the end-user very little flexibility in crafting the *expressive* domain for training the explainer. While [1,2] took a step in the direction of incorporating constraints in analyzing ML models, their approaches are (1) only applicable for verification tasks, (2) operate on very specific ML models (BNNs), and (3) are not model-agnostic. To the best of our knowledge, CLIME is the first work to propose the use of constraints in a principled and model-agnostic way, significantly extending the capabilities of existing work including LIME.
> > >
> > > Secondly, we agree with the reviewer that it may be possible to enhance LIME with simple constraints using rejection sampling. However, this approach may not scale in general. Concretely, the size of the constrained space can be very small relative to the larger domain that LIME samples from, resulting in most samples getting rejected. This can make LIME's perturbation procedure extremely inefficient to the point of being infeasible. This situation is not uncommon; for instance it occurs in the adversarial robustness experiments. Consider the problem of generating instances that are at Hamming distance 2 from a given instance over 20 Boolean features. In expectation, only 1 sample would be accepted for every 5500 rejections and this quantity increases exponentially with number of features.
> > >
> > > [1] Nina Narodytska, Aditya A. Shrotri, Kuldeep S. Meel, Alexey Ignatiev, and João Marques-Silva. Assessing heuristic machine learning explanations with model counting. In Theory and Applications of Satisfiability Testing - SAT 2019 - 22nd International Conference, SAT 2019, Lisbon, Portugal, July 9-12, 2019, Proceedings
> > > [2] Shih, Andy, Adnan Darwiche, and Arthur Choi. "Verifying binarized neural networks by angluin-style learning." International Conference on Theory and Applications of Satisfiability Testing. Springer, Cham, 2019.
> > >
> > >
> > > **Q2.1: [Ease of Sampling]**
> > > The reviewer pointed that "a primary issue with neighborhoods in LIME is not that they are difficult to sample from". We would like to clarify this point as performing the sampling with guarantees is actually a difficult problem, both theoretically and practically, [3,4].
> > >
> > > Indeed, it is easy to come up with a lightweight and simple sampling procedure such as LIME's. However, this comes with a sacrifice of correctness. For instance, LIME's simple procedure is unable to take advantage of the weight function during the sampling phase. As a compromise, LIME samples neighboring instances without taking the weights into account, and reweights them later when training the explainer model.  While, it might look as a good compromise initially, problematic issues immediately manifest as we challenge LIME as in Slack et al.
> > >
> > > [3] Valiant, Leslie G. "The complexity of computing the permanent." Theoretical computer science 8.2 (1979): 189-201.
> > > [4] Jerrum, Mark R., Leslie G. Valiant, and Vijay V. Vazirani. "Random generation of combinatorial structures from a uniform distribution." Theoretical computer science 43 (1986): 169-188
> > >
> > > **Q2.2: [How realistic is to expect a practitioner to write down logical constraints before querying explanations?]**
> > > Our work enhances LIME with powerful capabilities. We admit that our framework does assume that the user has some understanding of the system and is able to communicate their knowledge via a formal constraint language. However, we would like to emphasize that many tools/techniques that deal with formal analysis of systems, programs, ML models, etc have similar expectations from users. For example, given the wide industrial adoption of formal verification toolkits where the user is also expected to specify system's properties, we strongly believe that enhancing a popular explainer with the ability to incorporate user-defined constraints can be useful in many domains.
> > >
> > > In the paper, we demonstrated two use-cases of our framework. For refining explanations, defining constraints is straightforward; the user just projects on different subspaces. We also observed that it was easy to come up with constraints to mitigate adversarial attacks by Slack et al. Unfortunately, running a user study with domain experts to quantify usability w.r.t. defining constraints is beyond the scope of the paper.

---

### Official Review · AnonReviewer2 · 2020-10-27

**Rating:** 6
**Confidence:** 3

**Review:**

The paper presents augmentations to the LIME explanation system using logical constraints. Given a black-box classifier f, LIME generates local explanations of an input x by sampling in the neighborhood of x, and learning a linear classifier that matches f in that neighborhood. The paper claims that this procedure is problematic because the samples in the neighborhood of x can be out-of-distribution (OOD), so learning from the OOD samples are not useful. The paper also claims that LIME cannot work in constrained spaces, although this is a similar point to OOD samples. Lastly, the paper claims to certify the estimation quality more quickly  by breaking early if the quality is below a certain threshold.

The problem of generating explanations is important, and the use of logical constraints to focus the attention of the explanation generation is interesting. However, the contributions of the paper are unclear, and the experimental results were not very exciting.

The paper seems to present two contributions: 1) generating explanations given a constraint and 2) certifying quality of explanations. But for 1) it seems they are simply calling off-the-shelf tools that sample from a logical constraint. It is unclear to me if there were any deeper insights than that. For 2), their main claim of improvement comes from the assumption that it is okay to terminate early if the estimated fidelity is small. This is not an unreasonable assumption, but the contribution is again unclear -- the paper is simply targeting an easier estimation problem (and existing approaches can probably do much better if they also make this assumption).

Since I didn't notice any deeper technical contributions, I was hoping to be convinced by experimental results. After just reading the introduction, my top concern was -- how realistic is it to generate logical constraints for an input x that you want an explanation for? The intro tried to motivate it as doctors setting constraints on features of a patient, so I was hoping for a real-world motivated scenario for constraining the explanation space in the experiments, and some measure of how the CLIME explanations are better than those of LIME. Unfortunately, the constrained spaces for Recividism doesn't seem to be motivated at all, and I don't see why I should believe that CLIME explanations are better than LIME just based on Fig1.

If I were to reshape the paper, I would spend much less space on the technical contributions, as the insights are fairly straightforward and can be described very quickly. I think the main concern (that I still think is not addressed) is how realistic / easy it is to expect a practitioner to write down logical constraints before querying explanations for a given input x. I think there is a big leap of faith here, that needs to be addressed experimentally via real-world case studies (e.g. doctor example in the introduction) where logical constraints are very natural to come up with, and that explanations generated using these logical constraints are much better.

---

> ### Author Response · Authors · 2020-11-13
> **Writing Constraints in Practice**
>
>
>
> **Q3: [How realistic is to expect a practitioner to write down logical constraints before querying explanations?]**
> Our work enhances LIME with powerful capabilities. We admit that our framework does assume that the user has some understanding of the system and is able to communicate their knowledge via a formal constraint language. However, we would like to emphasize that many tools/techniques that deal with formal analysis of systems, programs, ML models, etc have similar expectations from users. For example, given the wide industrial adoption of formal verification toolkits where the user is also expected to specify system's properties, we strongly believe that enhancing a popular explainer with the ability to incorporate user-defined constraints can be useful in many domains.
>
> In the paper, we demonstrated two use-cases of our framework. For refining explanations, defining constraints is straightforward; the user just projects on different subspaces. We also observed that it was easy to come up with constraints to mitigate adversarial attacks by Slack et al. Unfortunately, running a user study with domain experts to quantify usability w.r.t. defining constraints is beyond the scope of the paper.

---

> ### Author Response · Authors · 2020-11-13
> **Contributions, Comparison to LIME, Other Approaches**
>
> Thank you for your feedback. We will upload an updated version of the paper in few days taking into account your comments and suggestions.
>
> [**Important clarification about the contributions**]
>
> Our work is tackling known drawbacks of the widely used explainer LIME: OOD-sampling and lack of fidelity guarantees for the explanations. To the best of our knowledge, these problems are still open and our work proposes an effective solution toward mitigating these issues. Therefore, we believe that our contribution is significant.
>
> **Q1: [contribution in "generating explanations given a constraint".]**
> We believe that the idea of enhancing LIME with user-defined constraints is novel. While, it might look simplistic in the retrospect, the benefits of CLIME are not apparent apriori. We demonstrated that using constraints helps (a) to mitigate the OOD-sampling problem and (b) to drill down and refine explanations, both of which are significant and powerful extensions of LIME's capabilities.
>
> Indeed, the sampling procedure uses an existing sampler and it leverages decades of research in the SAT community. Note that CLIME procedure for generating data for training the explainer model subsumes LIME's ad-hoc perturbation procedure as a special case.  In our view, the fact that we are able to take advantage of the state-of-the-art sampling tool is an advantage of our approach: new improvements to the sampling tools can be directly incorporated into the CLIME framework.
>
> [**Important clarification about "whether we should believe that CLIME explanations are better than LIME explanations"**]
> As we pointed out, the benefits of using constraints are two-fold and each experiment demonstrates complementary capabilities:
> Case (a) to mitigate the OOD-sampling problem as demonstrated in the "Adversarial Robustness experiments"
> Case (b) to give the user the ability to drill down and refine explanations as demonstrated by the "Recidivism experiments".
>
> In Case (a), the explanations generated by LIME are clearly misleading as shown by Slack et al.
> CLIME explanations are robust to this type of attack and we believe it is fair to say that our explanations are of better quality.
>
> In Case (b), we cannot claim that our explanations are better or worse than LIME's.
> Our explanations provide additional insights on the classifier that LIME is unable to by design.
> Fig 1 demonstrates the kind of insights that CLIME is able to provide in addition to what is made available by LIME.
>
>
> **Q2: [contribution in "certifying quality of explanations".]**
> To the best of our knowledge, this is the first work on post-hoc, model-agnostic certification of quality of explanations. We believe that identifying the applicability of early-termination to the problem of quality certification is a significant insight. Moreover, the benefits of estimation framework go beyond early termination; it enables us to certify fidelity in a model-agnostic way (unlike Narodytska et al.) while also allowing approximate-sampling from the universe (unlike the original algorithm by Dagum et al.) (please see Sec. 4).
>
> [**Important Clarification about Other Approaches and Early-Termination**]
> It is indeed possible to terminate early when using alternative estimation approaches. However,
> it has been reported in the literature that these approaches would not benefit from early termination [Meel et al.]. We will be happy to discuss more as needed. We will present an empirical evaluation of Algorithms 2,3 in the updated version.
>
> [Narodystka et al.] Nina Narodytska, Aditya A. Shrotri, Kuldeep S. Meel, Alexey Ignatiev, and João Marques-Silva. Assessing heuristic machine learning explanations with model counting. In Theory and Applications of Satisfiability Testing - SAT 2019 - 22nd International Conference, SAT 2019, Lisbon, Portugal, July 9-12, 2019, Proceedings
> [Dagum et al] P. Dagum, R. Karp, M. Luby, and S. Ross. An optimal algorithm for Monte Carlo estimation. SIAM Journal on Computing, 29(5):1484–1496, 2000.
> [Meel et al.] Kuldeep S Meel, Aditya A Shrotri, and Moshe Y Vardi. Not all fprass are equal: demystifying fprass for dnf-counting. Constraints, 24(3-4):211–233, 2019.

---

> > ### Comment · AnonReviewer2 · 2020-11-17
> > **Response**
> >
> > Thanks for your response, I appreciate the clarifications. I'm considering the paper more favorably now (score updated).
> >
> > Just a couple of points:
> > 1) I'm not a fan of Figure 1 -- as a reader its hard to tell what the takeaway message is. Some of the text in the main paper should go in the caption, and the relevant bars (race, married, alcohol) should be highlighted visually.
> >
> > 2) **it has been reported in the literature that these approaches would not benefit from early termination**
> > Can you point to where they report this in the Meel paper? I think this is a rather important point, and should definitely be clarified in the main paper. Currently it says "but require far too many samples when ρ is low" but that sounds like if they terminate early their method will be just as fast too.
> >
> > I am also interested in the experiments for Alg 2,3 that Reviewer5 suggested.

---

> > > ### Author Response · Authors · 2020-11-24
> > > **Early Termination**
> > >
> > > Thank you for your comments.
> > >
> > > Q1. **[Clarity of Figs comparing LIME and CLIME explanations]**
> > > Thank you for your suggestions on how to improve the figure comparing LIME and CLIME explanations on Recidivism Dataset. We have highlighted the relevant features and improved readability.
> > >
> > > Q2. **[Clarification on Benefits of early termination]**
> > > The Meel paper compares two broad classes of estimation algorithms: (1) Algorithms based on Monte Carlo Sampling which includes the approach we use in the current paper (2) Algorithms that use Hash functions eg: ApproxMC [Soos et al.] which were used by [Narodytska et al.]
> > >
> > > A discussion on scalability of the two classes of algorithms wrt the size of the quantity being estimated is presented in detail in Section 7 of the Meel et al. paper. Hashing-based approaches scale well for estimation problems where the quantity being estimated is very small relative to the size of the universe. In fact, it is infeasible to use Monte Carlo approaches in practice for such problems, as the number of samples required to get a good estimate can grow exponentially large. Naturally, for hashing-based approaches, there would be *no benefit from terminating early when the quantity being estimated is small, as they are already faster than Monte Carlo based approaches in this regime*. On the other hand, for problems where the quantity being estimated is a large fraction of the size of the universe, Monte Carlo approaches require only a handful of samples to generate a good estimate, and hence are extremely fast as compared to hashing-based techniques. Note that the need for two-sided bounds *precludes* early termination in this regime.
> > >
> > > In order to test our scalability hypothesis in practice, we evaluated the efficiency of our algorithm on benchmarks for estimating fidelity of Anchor explanations. These benchmarks were used in Narodytska et al. in conjunction with the hashing-based tool (ApproxMC). In Fig. 2, we conclusively show the superior scalability of our tool as compared to the hashing-based tool (additional experiments and results in Appendix). This is in line with what was predicted by the Meel et al. paper.
> > >
> > > [Soos et al.] Soos, Mate, Stephan Gocht, and Kuldeep S. Meel. "Tinted, Detached, and Lazy CNF-XOR Solving and Its Applications to Counting and Sampling." International Conference on Computer Aided Verification. Springer, Cham, 2020.
> > >
> > > [Narodystka et al.] Nina Narodytska, Aditya A. Shrotri, Kuldeep S. Meel, Alexey Ignatiev, and João Marques-Silva. Assessing heuristic machine learning explanations with model counting. In Theory and Applications of Satisfiability Testing - SAT 2019 - 22nd International Conference, SAT 2019, Lisbon, Portugal, July 9-12, 2019, Proceedings
> > >
> > > [Meel et al.] Kuldeep S Meel, Aditya A Shrotri, and Moshe Y Vardi. Not all fprass are equal: demystifying fprass for dnf-counting. Constraints, 24(3-4):211–233, 2019.

---

### Official Review · AnonReviewer3 · 2020-10-28
**Fixing problems caused by out-of-distribution sampling in LIME explanations using Boolean constraints**

**Rating:** 7
**Confidence:** 3

**Review:**

The topic of this paper is highly topical, as it focuses on providing explanations for black-box classifiers. The paper tackles the known issue with out-of-distribution sampling in Ribeiro et al.'s LIME. The approach taken is to introduce the use of Boolean constraints in LIME algorithm, which allows for providing some guarantees on the quality of the explanations.

The paper is generally well-written and the level of details provided makes it a pleasure to read. However, a bit more details could be provided at places.

Related work is sufficiently covered in the main text, and more discussion is provided in the appendix. The authors might take a look to a recent work Bjorlund et al. that also tries to remedy the out-of-distribution sampling problem using an approach based on robust regression.

[Sparse robust regression for explaining classifiers
A Björklund, A Henelius, E Oikarinen, K Kallonen, K Puolamäki
DS 2019: Discovery Science, 351-366]

The experimental evaluation (in the main paper) is brief, missing, e.g., much of the details to describe the experimental setting used. Since the approach taken is to use Boolean constraints, it would be relevant to know the effort needed to perform the sampling, i.e., does the runtime of CLIME differ much of to that of LIME.

Pros:
1. Very relevant topic (explanations) and demonstration of the importance and the relevance of the data distribution in explaining black-box classifiers
2. Analysis and proven guarantees of explanation quality
3. Generally well written paper and clear scope, with a more detailed descriptions provided in the appendices.

Cons:
1. Rather much of the relevant material is in the appendices
2. Not enough details of the experimental evaluation provided
3. The end result (the actual explanations and how a human would understand them) is only very briefly touched

Questions:

If I understand correctly, CLIME can still introduce samples that are not necessarily from the distribution of the original data (which might be totally unknown, and the original data would be the only information available), as it seems to me that the difference to LIME is that the samples need to obey the Boolean constraints introduced. Is this correct?

Would this then not mean that the "correctness" of the explanation heavily relies on using "correct" Boolean constraints in the sampling procedure? Could there be a scenario, where the Boolean constraints used in the sampling process would lead to meaningless results, i.e., the samples used would be outside the data space for the (unknown) distribution of the data?

The work covers  explanations for (binary) classifiers and the focus is on tabular data. Do you see any (e.g., performance) issues that could affect the usability of CLIME in a more general scenario?

What is the performance of the sampling algorithm, i.e., how does CLIME compare to LIME when considering the time?

Could you provide more details of the experimental evaluation, i.e., details of the classifiers trained, training / testing dataset splits, performance, will the code be made openly available?

Minor details:

Some inconsistencies in the algorithms:
Alg. 1 is missing input (only provided in the main test) which makes the code hard to follow
Alg. 1 and Alg. 3 both call getSamples, but have different number of arguments

Fig. 1 the labels in x-axis are way too small.

MLP undefined in page 7

Also, page 7, "that gives 69.1%" (missing "accuracy"?)

Page 7, the sentence "e.g. if the height of the bar is 0.1 the corresponding feature is mostly top ranked in all explanations." is quite unclear, since the smallest values in Fig 1 are roughly 1

Check the capitalisation in the titles in References (e.g., bayes, dnf, shap, lime, bayesian should not be small caps)

---

> ### Author Response · Authors · 2020-11-13
> **Sampling differences, Performance, Experiment Details**
>
> Thank you for your feedback and the pointer to the missing reference. We will upload an updated version of the paper in a few days taking into account your comments and suggestions.
>
> Answers to Questions:
>
> **Q1,2: [Differences between LIME and CLIME sampling]**
> Indeed, one difference to LIME is that all the samples obey the Boolean constraints in CLIME. While the enhancement of LIME with user-defined constraints is a powerful feature of our framework, we agree that it can be misused by the user, e.g. Boolean constraints might  incorrectly focus on OOD inputs sub-space.
> However, it is often fairly straightforward to express in-distribution subspaces through constraints. For example, even simple Hamming distance constraints were sufficient for capturing in-distribution samples in Sec. 5. We emphasize that the original LIME framework doesn't support such capabilities. We discuss other benefits of CLIME in the replies to Reviews 3 and 4.
>
> **Q3,4: [Performance of CLIME compared to LIME]**
> Uniform (or weighted) sampling is a hard problem with a high complexity (it is reducible to the problem of counting solutions which is #P-Complete). Hence, one should expect some performance degradation when compared to LIME's simplistic perturbation procedure. A thorough evaluation of CLIME's run-time performance on different data-types is beyond the scope of this paper as it involves extensive consideration of various tunable parameters of the sampling tools along with SAT encodings.
>
> For the use-case of refining explanations, constraints are likely to be rather simple, e.g. when projecting over input sub-spaces. In this case, CLIME's performance is very close to LIME's performance. If constraints are more involved, like Hamming distance constraints, sampling takes longer but is still reasonable e.g. it took less than 2 hours to generate the data for Fig 2A i.e. 200 explanations for each of the 5 hamming distances, with 5000 samples per explanation (5*10^6 samples in total) on a standard laptop.
>
> **Q5: [Additional experimental details and code]**
> We will include a link to the code repository in the final version of the paper. Appendices D.3.2 and D.4 include a detailed discussion of the experimental setup for the adversarial robustness experiments. We will use the additional page for moving some information from the appendix to the main text.
> Finally, we will have add experiments for Algorithms 2,3 in the next updated version.

---

### Official Review · AnonReviewer5 · 2020-11-05

**Rating:** 6
**Confidence:** 4

**Review:**

This paper presents a method --- CLIME --- to generate constrained explanations using LIME.  The method relies on boolean constraints to dictate the sampling region for LIME. This approach offers a number of advantages.  First, users can compare important features for groups conditioned on different factors. Second, the method adds a level of robustness against adversarial attacks.  The authors demonstrate their approach on a number of data sets and find useful insights conditioning on different features and greater robustness to attacks proposed by Slack et. al.

Questions + Comments for the authors:
- The authors consistently reference motivating examples left for the appendix throughout the paper.  These are phrased as important examples are discussed in detail in section 4.1.  Please consider using the space afforded by the revision to introduce these in the main text because this currently affects readability.
- Is there any empirical verification for the methods provided in 4.2 (i.e. algorithms 2 & 3)? I understand the result is phrased as a theoretical result, but there is a lot of attention devoted to it in the paper and it would be good to evaluate it in practice.  It would be nice to demonstrate the output of this estimate for an example and provide some analysis there (unless I'm missing something?)
-  The extension of the attacks to only discrete data for LIME is interesting -- making it much more similar to the SHAP attack in some sense because this attack only relies on discrete data. I'm a bit confused about the claim that  the methods help detect adversarial attacks.  If they're using the adversarial classifier from Slack et. al., shouldn't the method in 2b reveal that the underlying classifier is only relying on the Race feature?  Could you clarify how this reveals the method detects the attacks?

Overall, the constrained explanation approach seems pretty useful for conditioning explanations and could be a valuable contribution.  It also seems pretty straightforward to implement on top of lime (not a bad thing) so could be of immediate value.

What I'm most confused about right now is (1) evaluation for section 4.2.  I see this as a useful result because the number of perturbations are a big issue for lime and this could be useful to assess the fidelity of explanation with more confidence + have more guidance if the fidelity is trustworthy. I'd be interested to at least see fidelity plotted against delta for an example or something like this.  Next, I'm struggling to see how this method reveals the Slack et. al. attacks.  As, I said, I'd expect to see a method which is not fooled by them to put race as the most important feature.  I'm currently leaning weak accept because I do see value in the method but would appreciate a small empirical extension to the results in section 4.2 and clarification around how the method reveals adversarial attacks.

---

> ### Author Response · Authors · 2020-11-13
> **Empirical Evaluation and Robustness vs Detection Clarification**
>
> Thank you for your suggestions for improving the paper. We will upload an updated version of the paper in a few days taking into account your comments and suggestions, e.g. we will move some examples to the main text.
>
> **Q1: [Effectiveness of Algorithms 2,3  compared to existing techniques]**
> We will present an empirical evaluation of Algorithms 2,3 in the updated version.
>
>
> **Q2: [Robustness vs detection of adversarial attacks from Slack et al using CLIME]**
> For an explainer that is not fooled by Slack et al., Race should indeed dominate the explanation. For CLIME, this is seen in the first column of Fig 2a when the Hamming distance is constrained to be small (h=2). This shows that enforcing the Hamming distance constraint is one way of making CLIME robust to adversarial attacks.
>
> Interestingly, CLIME's abilities go beyond robustness: we show how CLIME can be used to *detect* the adversarial nature of a classifier. We experimentally observed that for a non-adversarial classifier, the relative importance of a feature in an explanation remains roughly the same for different distances. For instance, in Fig. 2B, if the feature F is the most important feature for samples of Hamming distance = 2 from $x$, then F is still important for samples with Hamming distance = 4. In contrast for adversarially constructed classifier, we observed that this property no longer holds. For example in Fig. 2A, the dominance of Race goes down drastically with increasing Hamming distance, which hints that the classifier is compromised.  We will clarify that our method is based on an empirical observation that works well for attacks from  Slack et al. We admit that it might be possible to have an adversarial classifier where our approach is not effective. We will clarify our claims on the detection of adversarial attacks.

---

> > ### Comment · AnonReviewer5 · 2020-11-14
> > **Response**
> >
> > In regards to Q2, I see what's being shown here now.  Thanks for the clarification. I think this is an interesting result and demonstrates how this method could be used for defense against such attacks.
> >
> > I'm still curious to see the aforementioned experimental evaluation for algorithms 2/3 in the update and appreciate the description the authors have provided so far.

---

> > > ### Author Response · Authors · 2020-11-24
> > > **Additional experiments in the updated version**
> > >
> > > We have added experimental evaluation of our estimation framework in the updated version.

---

### Author Response · Authors · 2020-11-24
**Summary of Changes**

We thank the reviewers for their comments and suggestions! Here is a summary of the main changes in the updated version:

**[Additional experiments on scalability of Alg. 2,3]**
We evaluated the efficiency of our algorithm on benchmarks for estimating fidelity of Anchor explanations [from Narodytska et al]. We compared our technique to the approach of [Narodytska et al.] which employs the state-of-the-art hashing-based tool called ApproxMC.

Our results demonstrate:

1. Our approach is ~7.5 times faster compared to ApproxMC, for the same probabilistic guarantees.

2. In practice, the error in our estimates is much smaller than the allowed theoretical bounds.

3. Our approach scales better than ApproxMC when stronger guarantees on tolerance and confidence are desired (see Discussion, Appendix D.4).

In Fig. 2, we provide a summary plot of our experiments. It shows the superior scalability of our tool as compared to the hashing-based tool on all benchmarks (additional experiments and results in Appendix D.4)

[Narodystka et al.] Nina Narodytska, Aditya A. Shrotri, Kuldeep S. Meel, Alexey Ignatiev, and João Marques-Silva. Assessing heuristic machine learning explanations with model counting. In Theory and Applications of Satisfiability Testing - SAT 2019 - 22nd International Conference, SAT 2019, Lisbon, Portugal, July 9-12, 2019, Proceedings

**[Improvements of the paper]**
improved readability of the paper, e.g. moved some examples/experiments to the main text, improved the figure comparing LIME and CLIME explanations on recidivism, clarified strengths and limitations of our adversarial attack detection method, fixed typos.

**[Related work]**
Expanded related work, which included adding and discussing references that the reviewers pointed out.

---

### Decision · Program_Chairs · 2021-01-07
**Final Decision**

**Decision:**

Reject

**Comment:**

The authors present CLIME, a variant of LIME which samples from user-defined subspaces specified by Boolean constraints. One motivation is to address the OOD sampling issue in regular LIME. They introduce a metric to quantify the severity of this issue and demonstrate empirically that CLIME helps to address it. In order to stay close to the data distribution, they use constraints based on Hamming distance to data points. They demonstrate that this approach helps to defend against the recent approach of Slack et al. 2020 to fool LIME explanations.

The paper is close to borderline, though concerns remain about experimental validation and the extent of novel contribution, since the original LIME framework is more flexible than described here and allows a custom distance function. Rev 1 believes that the original LIME framework is sufficient to handle Hamming distance constraints though sampling will be less efficient. To their credit, authors engaged in discussion but this should be further elaborated in a revised version.